**Measuring Light Absorption by Freshly Emitted Organic Aerosols: Optical Artifacts in**
**Traditional Solvent Extraction-Based Methods**
Nishit J Shetty[1], Apoorva Pandey[1], Stephen Baker[2], Wei Min Hao[2], Rajan K. Chakrabarty[1,3]
[1]Center for Aerosol Science and Engineering, Department of Energy, Environmental and Chemical
Engineering, Washington University in St. Louis, St. Louis, MO 63130, USA
[2]USDA Forest Service, Rocky Mountain Research Station, Fire Sciences Laboratory, Missoula, Montana,
USA
[3]McDonnell Center for the Space Sciences, Washington University in St. Louis, St. Louis, MO 63130,
USA
*Correspondence to*: Rajan K. Chakrabarty (chakrabarty@wustl.edu)
**Abstract**
Recent studies have shown that organic aerosol (OA) could have a non-trivial role in atmospheric
light absorption at shorter visible wavelengths. Good estimates of OA light absorption are
therefore necessary to better estimate radiative forcing due to these aerosols in climate models.
One of the common techniques used to measure OA light absorption is the solvent extraction
technique from filter samples which involves the use of a spectrophotometer to measure bulk
absorbance by the solvent-soluble organic fraction of particulate matter. Measured solvent phase
absorbance is subsequently converted to particle-phase absorption coefficient using scaling
factors. The conventional view is to apply a correction factor of 2 to absorption coefficients
obtained from solvent-extracted OA based on Mie calculations. The appropriate scaling factors are
a function of biases due to incomplete extraction of OC by solvents and size-dependent absorption
properties of OA. The range for these biases along with their potential dependence on burn
conditions is an unexplored area of research.
Here, we performed a comprehensive laboratory study involving three solvents (water, methanol,
and acetone) to investigate the bias in absorption coefficients obtained from solvent extraction-
based photometry techniques as compared to in-situ particle phase absorption for freshly emitted
OA from biomass burning. We correlated the bias with OC/TC mass ratio and single scattering
albedo (SSA) and observed that the conventionally used correction factor of 2 for water and
methanol-extracted OA might not be extensible to all systems and suggest caution while using
such correction factors to estimate particle-phase OA absorption coefficients. Furthermore, a linear
correlation between SSA and OC/TC ratio was also established. Finally, from the spectroscopic
data, we analyzed the differences in Absorption Ångström Exponents ($A\mathring{A}E$) obtained from
solution- and particulate-phase measurements. We noted that $A\mathring{A}E$ from solvent phase
measurements could deviate significantly from their OA counterparts.
**1  Introduction**
Carbonaceous aerosols constitute a major short-lived climate pollutant, and even though they have
been studied extensively in recent years, estimates of their contribution to shortwave radiative
forcing remains highly uncertain (IPCC, 2013). Based on their thermal-refractory properties,
carbonaceous aerosols are categorized as elemental carbon (EC) or organic carbon (OC) (Chow et
al., 2007b; Bond et al., 2013), and the sum of OC and EC is referred to as total carbon (TC). When
defined optically, the refractory EC component is approximately referred to as black carbon (BC)
(Chow et al., 2007b; Bond et al., 2013). BC aerosol constitute the strongest of the light absorbing
aerosol components in the atmosphere (Ramanathan and Carmichael, 2008; Andreae and
Gelencsér, 2006; IPCC, 2013). While BC absorbs strongly in the visible spectrum, the contribution
of OC towards absorption has largely been neglected, even though many studies have
demonstrated significant OC absorption at lower visible wavelengths (Yang et al., 2009; Chen and
Bond, 2010; Chakrabarty et al., 2010; Kirchstetter 2012). The atmospheric mass of OC can be 3-
12 times larger than that of BC (Husain et al., 2007; Zhang et al., 2008) which warrants its inclusion
as an atmospheric light absorber. Only recently have global modeling studies started incorporating
radiative forcing by organic aerosol (OA) absorption (Wang et al., 2014; Saleh et al., 2015; Lin et
al., 2014; Wang et al., 2018). Thus, having accurate estimates for OA absorption is necessary to
help improve climate models.
A convenient and prevalent methodology of measuring OA absorption is based on collecting
aerosol particles on a filter substrate followed by extracting the organic compounds into a solvent.
This analytical method is used in many studies as it ideally excludes any interference from EC and
primarily provides the absorption spectra of extracted OC (Mo et al., 2017; Chen and Bond, 2010;
Liu et al., 2013). The absorbance of organic chromophores in the solvent extract is measured using
an ultraviolet-visible (UV-Vis) spectrophotometer and measured absorbance values can be
converted to corresponding solvent phase absorption coefficients ($b_{abs,sol}$). However, this
methodology has limitations as it is unable to represent size-dependent absorption properties of
the extracted OA (Liu et al., 2013; Washenfelder at al., 2015; Moosmüller et al., 2011). To correct
for this limitation, the complex refractive index (RI) of OC is estimated by assuming the real part
and calculating the imaginary part for extracted OC using $b_{abs,sol}$ and dissolved OC concentration,
the complex RI is then used along with a number size distribution as inputs to Mie theory for
calculating the particle-phase absorption coefficient for dissolved OC. In addition to discrepancies
between particle and solvent phase optical properties, the method suffers from biases due to
incomplete extraction of organics by different solvents (Chen and Bond, 2010; Liu et al., 2013)
which lead to differences in values of $b_{abs,sol}$ obtained from different solvents. The significance
and extent of this bias varies based on the OC extraction efficiency of a given solvent and would
be negligible for solvents extracting 100% of organic chromophores. A combination of inefficient
organic carbon extraction and the methods inability to measure size-dependent OA absorption
properties can result in significant errors to optical properties obtained using this method. Despite
the low OC extraction efficiency of water (Chen and Bond, 2010) and large potential for errors,
past studies have used light absorption by water soluble organic carbon (WSOC) as a surrogate for
OA optical properties (Bosch et al., 2014; Kirillova et al., 2014a; Kirillova et al., 2014b). However,
the use of water as an OA surrogate is decreasing with more recent studies using methanol to
extract OC (Cheng et al., 2016; Shen at al., 2017; Xie et al., 2017). While methanol has a higher
OC extraction efficiency than water (Chen and Bond, 2010), its efficiency is limited ranging from
85-98% (Cheng et al., 2016; Xie et al., 2017) which can lead to misrepresentation of OA optical
properties if the unextracted fraction correspond to extremely low volatility organic carbon
(ELVOCs) or similar organic chromophores which have large light absorption efficiencies (Saleh
et al., 2014), underscoring the need for a more complete extraction protocol. In addition to
problems with incomplete OC extraction, previous studies have attempted to correct for size-
dependent biases using absorption coefficients determined with Mie theory and provided a narrow
range of solvent-dependent scaling factors from 2 for water extracts to 1.8 for methanol extracts,
all corresponding to a mean particle diameter of 0.5 μm (Liu et al., 2013; Liu et al., 2016;
Washenfelder et al., 2015). Sun et al. (2007) performed theoretical calculations and postulated a
correction range of 0.69 - 0.75 for OC particles with diameters much smaller than the wavelength
of light. These correction factors while applicable to these individual systems, might not be
extensible to aerosol emissions from other combustion events. However, many studies have used
scaling factors from such studies on absorption coefficients obtained from solvent phase optical
measurements despite potential differences in system dependent biases for each experiment (Kim
et al., 2016; Zhang et al., 2017; Wang et al., 2018). To the authors knowledge, no attempts have
been made to explicitly study or quantify these biases with varying aerosol intrinsic properties,
such as the EC/OC ratios, and single scattering albedo (SSA), even though these properties have
shown to be well correlated with OA optical properties (Zhang et al., 2013; Saleh at al., 2014;
Bergstrom et al., 2007).
In-situ measurement of particulate-phase absorption coefficient is commonly and accurately
accomplished using a photoacoustic spectrometer (PAS) (Lack et al., 2006; Arnott et al., 2005;
Arnott et al., 2003). However, on its own, a single-wavelength PAS cannot distinguish between
absorption by OC and BC aerosol and it typically measures the total particle-phase absorption
coefficient ($b_{abs,tot}$) of the aerosol population in the cell (Moosmüller et al., 2009). One can make
use of a multi-wavelength PAS using which the OA absorption coefficient ($b_{abs,OA}$) could be
separated out from that of BC absorption, based on the difference in BC and OA Absorption
Ångström Exponent ($A\mathring{A}E$) (Washenfelder et al., 2015; Arola et al., 2011; Kirchstetter and
Thatcher; 2012). The $A\mathring{A}E$ for pure BC is well-constrained at 1 in the visible and near-infrared
wavelengths (Moosmüller et al., 2009). The value of $b_{abs,OA}$ is calculated as the difference between
$b_{abs,tot}$ and the BC absorption coefficient. A possible technique to measure the bias between
particle and solvent phase organic absorption ($b_{abs,OA}/b_{abs,sol}$) can thus be established by carrying
out simultaneous measurements of  solution- and particle-phase absorption properties during a
study. Determining $b_{abs,OA}$ using this method gives large errors when BC absorption coefficient
is large or comparable to $b_{abs,tot}$ as $b_{abs,OA}$ would be a small number obtained by the subtraction
of two large numbers limiting the use of this technique for relatively low EC/OC ratios.
Here, we burnt a range of different biomass fuels under different combustion conditions and the
resulting aerosol emissions were passed through various in-situ instruments while simultaneously
being collected on quartz-fiber filters. The particle phase absorption coefficient was obtained using
integrated photoacoustic-nephelometer spectrometers (IPNs) at wavelengths 375, 405 and 1047
nm. Organics collected on quartz-fiber filters were extracted in water, acetone, and methanol, and
corresponding $b_{abs,sol}$ values were calculated. These values were compared with corresponding
$b_{abs,OA}$, and the change in $b_{abs,OA}/b_{abs,sol}$ with varying single scattering albedo (SSA) values and
OC/TC ratios was examined. SSA was parametrized with the OC/TC ratios with trends similar to
those observed by Pokhrel et al. (2016). $A\mathring{A}E$ from spectroscopic data for solution and particle
phase measurements were compared, and the Mie Theory based correction factor was also
investigated for a few samples.
**2   Methods:**
**2.1   Sample generation and collection**
Fig. 1 is a schematic diagram of our experimental setup, which consisted of a sealed 21 m$^3$
stainless-steel combustion chamber housing a fan for mixing and recirculation (Sumlin et al.,
2018b). Aerosol samples were generated by burning several types of biomass including pine, fir,
grass, sage, and cattle dung (details are provided in the Supplementary Information). During a
chamber burn, 10-50 g of a given biomass was placed in a stainless-steel pan and ignited by a
butane lighter. The chamber exhaust was kept closed for the duration of a given experiment. The
biomass bed was either allowed to burn to completion or it was prematurely extinguished and
brought to a smoldering phase by extinguishing the flame beneath a lid. Different combustion
conditions were used to generate samples with varying properties: OC/TC ratios ranged from 0.55-
1, and SSA values ranged from 0.56-0.98 for wavelengths of 375, 405, and 1047 nm.
For one set of experiments, the particles were directly sampled from the chamber; in another set,
the sampling was done from a hood placed over the burning biomass. A diffusion dryer removed
excess water from the sample stream, and the gas-phase organics were removed by a pair of
activated parallel-plate semi-volatile organic carbon (SVOC) denuders. The gas-phase organics
were stripped to reduce artifacts produced by the adsorption of organic vapors on the quartz filters.
The aerosols were finally sent to a 208-liter stainless-steel barrel, from which they were
continuously sampled by the three IPNs. Some phase repartitioning of condensed SVOC into the
vapor phase might have taken place post-denuding in our holding tank and could have introduced
a positive bias to our filter-based measurements. The experiments were conducted in two sets, the
first set included a scanning mobility particle sizer (SMPS, TSI, Inc.) and size measurements from
this instrument were used in Mie Theory calculations detailed in Section 2.3. The SMPS was not
used in the second set of experiments due to problems with aerosol flows in the system. However,
the SMPS data from the first set of experiments gave us an estimate of the range over which the
size distributions varied and was used to obtain the geometric mean of the size distribution. The
real-time absorption and scattering coefficients were measured by the IPNs, and samples were
simultaneously collected on quartz fiber filters once a steady state signal was achieved. The
absorption and scattering coefficients were used to calculate the SSA, which is simply the
scattering coefficient divided by the extinction coefficient. Radiative forcing calculations for
absorbing OC require good estimates of OC absorption at different SSA values (Lin et al, 2014;
Feng et al, 2013; Chakrabarty et al, 2010) underscoring the need to study OA absorption biases as
a function of SSA. The particles were passed through the filter samplers at a flowrate of 5 L min⁻
[1], with sampling times ranging from 2-15 minutes. Two or more filters were collected for a given
steady state condition. One of these filters was used to determine the OC and EC fractions of the
deposited particles, and the other filters were used for the extraction experiments. The only
exception was one sample from dung combustion that we assumed to be purely organic aerosol
based on the smoldering only nature of the burn and previously analyzed optical dataset of aerosol
from similar burn conditions.

## 2.2 Analytical techniques

### 2.2.1 Absorption by solvent extracted OC

Quartz filters (Pallflex Tissuquartz, 47 mm diameter) collected during sampling were split into
four quarters, and each quarter was extracted using either deionized water, acetone, hexane, or
methanol. The absorption by hexane extracts were low and prone to errors, so data for its extracts
were not analyzed. The filters were placed in 3-5 ml of the solvent for 24 hours. The filter was not
sonicated to reduce artifacts from mechanical dislodging of BC particles (Phillips and Smith, 2017)
and to avoid changes in chemical composition caused by acoustic cavitation (Mutzel et al., 2012).
The solvent volumes were measured both before and after the extraction and the differences
between the two measurements were within 8%. The extracts were then passed through syringe
filters with 0.22 μm pores to remove any suspended particles introduced during the extraction
process.
The light absorbance of the extracts was measured using a UV-Vis spectrophotometer (Varian
Inc., Cary 50) at wavelengths from 300 nm to 800 nm. To compare the absorbance ($A(\lambda)$) of
chromophores in the solution with the absorption coefficient of the particles in the atmosphere, all
absorbance values were converted to solution-phase absorption coefficients at given wavelengths
$(b_{abs,sol}(\lambda))$ (Liu et al., 2013):
$$b_{abs,sol}(\lambda) = (A(\lambda) - A(700))\frac{V_l}{V_a * l} . \ln(10), \tag{1}$$

where $V_l$ is the volume of solvent the filter was extracted into, $V_a$ is the volume of air passed over
the given filter area, and $l$ is the optical path length that the beam traveled through the cuvette (1
cm). Absorbance at a given wavelength is normalized to absorbance at 700 nm to account for any
signal drift within the instrument. Absorbance at 700 nm was negligible and close to zero for the
analyzed samples indicating no absorption at long wavelengths and little to no signal drift for the
instrument. The resulting absorption coefficient (m$^{-1}$) was multiplied by $\ln(10)$ to convert from log
base 10 (provided by the UV-Vis spectrophotometer) to natural log.
**2.2.2 Absorption by BC and OC in particle phase**
To estimate the BC absorption at 375 nm and 405 nm, the absorption data from the IPN operated
in the infrared regime at a wavelength of 1047 nm was converted to equivalent BC particulate
absorption at the near UV wavelengths using a BC absorption Ångström exponent $(A\text{\AA}E_{BC})$ value
of 1 (Kirchstetter et al., 2004; Andreae and Gelencsér, 2006). It was assumed that all the absorption
at 1047 nm could be attributed to BC aerosol (Bahadur et al., 2012). The BC light absorption
coefficient at shorter wavelengths $(b_{abs,BC}(\lambda))$ was calculated by:
$$b_{abs,BC}(\lambda_1) = b_{abs,tot}(1047) . \left(\frac{\lambda_1}{1047}\right)^{-A\text{\AA}E_{BC}}, \tag{2}$$

where $\lambda_1$ is the wavelength at which the absorption will be calculated and $A\text{\AA}E$ is defined for a
pair of wavelengths $\lambda_1$, $\lambda_2$ as the exponent in a power law expressing the ratio of the absorption
coefficients as follows (Moosmüller et al., 2009):
$$A\mathring{A}E(\lambda_1\lambda_2) = \frac{ln\left[b_{abs}(\lambda_1)\big/b_{abs}(\lambda_2)\right]}{ln\left[\lambda_2\big/\lambda_1\right]}$$ (3)
$A\mathring{A}E$ is an optical descriptor of the inherent material property. For BC particles, typical values of
$A\mathring{A}E \approx 1$, while for OC particles $A\mathring{A}E > 4$ (Moosmüller et al., 2009). The value of $b_{abs,BC}$ at 375nm
and 405nm was then subtracted from $b_{abs,tot}$ at those wavelengths to calculate $b_{abs,OA}$. The ratio
$b_{abs,OA}/b_{abs,sol}$ was calculated to represent the scaling bias between the bulk solvent phase
absorption coefficient and OA absorption coefficient.
The organic and elemental carbon compositions of the filters were measured with a thermal-optical
OC/EC analyzer (Sunset Laboratory, Tigard, OR) using the Interagency Monitoring of Protected
Visual Environments (IMPROVE)-A Thermal/Optical Reflectance (TOR) analysis method (Chow
et al., 2007a). The OC/TC ratios were assumed to be constant for a given steady state IPN reading,
which allowed us to relate the absorption data to the OC/TC data. The assumption was tested by
performing EC/OC analysis of two filters collected during a given steady state for a burn. The
OC/TC ratio remained unchanged or within experimental error for the burns and results for the
EC/OC analysis of tested filters are provided in Table S1 and S2 of the Supplementary Information.
**2.2.3 Uncertainty using Monte Carlo simulations**
The uncertainties due to error propagation were evaluated using a Monte Carlo approach. The true
measurement value was assumed to possess a Gaussian probability distribution with the mean and
standard deviation corresponding to measured values and errors associated with the instrument
(Table S4), respectively. Calculations were performed by randomly selecting values based on the
probability distribution for the different variables and corresponding values for $b_{abs,OA}/b_{abs,sol}$
were estimated. A total of N = 10000 iterations were performed for each data point and each
simulation was rerun 100 times till the $b_{abs,OA}/b_{abs,sol}$ value converged for the calculations. The
propagated error due to uncertainty in important variables was then calculated as the standard
deviation of $b_{abs,OA}/b_{abs,sol}$ values acquired over simulations. A pseudocode for the Monte Carlo
calculation is detailed in the Supplementary Information along with Table S4 which denotes
typical mean and standard deviation values used for variables with uncertainties.
**2.3 Mie theory calculations**
A commonly used method to correct for differences between the chromophore absorption in
solution and aerosol particle absorption is by using Mie Theory (Liu et al., 2013; Washenfelder et
al., 2015). The imaginary part (*k*) of the complex refractive index *m = n + ik* can be determined
from bulk solution phase absorption data and converted to equivalent OA absorption using Mie
Theory along with assumptions regarding the shape of the particles and the real part of the complex
refractive index of the particle.
To find *k*, the mass absorption efficiency ($\alpha/\rho$) was determined using the absorbance data and the
OC mass concentration in the solution (Liu et al., 2013):
$$\frac{\alpha(\lambda)}{\rho} = \frac{b_{abs,sol}(\lambda)}{M},$$ (4)
where $b_{abs,sol}(\lambda)$ is the solvent-phase absorption coefficient determined in Eq. (1), and M is the
mass concentration of OC in the solution. In the given study, the OC mass concentration was
measured for some of the water extracts using a total organic carbon (TOC) analyzer (Shimadzu,
TOC-L). The water-soluble organic carbon (WSOC) was then used to estimate $\alpha/\rho$ of the solution.
The calculated $\alpha/\rho$ was further used to determine *k* for the WSOC by (Chen and Bond 2010):
$$k(\lambda) = \frac{\rho.\lambda.\left(\frac{\alpha(\lambda)}{\rho}\right)}{4\pi},$$ (5)
where $\lambda$ is the light wavelength at which $k$ needs to be calculated, and $\rho$ is the density of the
dissolved organic compounds. A $\rho$ value of 1.6 (Alexander et al., 2008) was used to calculate the
$k$ values, and was also used in all subsequent calculations using density. It is important to note that
$k$ values obtained using this method will represent optical characteristics of OC mass and not total
organic mass. A Mie based inversion algorithm was used to extract the real part of the refractive
index ($n$) using data from the SMPS and IPN (Sumlin et al., 2018a). If size distributions extended
over the SMPS measurement range, the data were extrapolated using a lognormal equation. A
sensitivity analysis was performed by varying the $n$ value from 1.4 to 2, and the change in Mie
calculated absorption was within 18%. The size distribution for the WSOC was estimated
assuming the same geometric mean and standard deviation as that of the original aerosol, but with
number concentrations calculated based on the extracted mass. Calculations for the number
concentration are provided in the Supplementary Information. After the size distribution and
complex refractive index were determined, they were used to calculate the absorption coefficient
based on Mie Theory, which was then compared to $b_{abs,sol}$ to verify the traditional Mie based
scaling factors for converting from solution to particle phase absorption.
**3 Results and discussion**
**3.1 Absorption bias correlated with single scattering albedo**
Fig. 2 shows the trends in $b_{abs,OA}/b_{abs,sol}$ for fresh organic aerosol emissions with varying SSA.
The different fuel types are marked with distinct markers and the error bars–accounting for
uncertainties in IPN, UV-Vis spectrophotometer, and extract-volume measurements, filter
sampling flowrates, and BC $A\mathring{A}E$ –are estimated from the results of the Monte Carlo simulation.
Measured SSA for pure fractal-like BC aggregates have values between 0.1-0.3 (Schnaiter et al.,
2003; Bond et al., 2013) depending on the size of the BC monomers (Sorensen 2001), and due to
this particularly low SSA of BC compared to OC, an increase in BC content of aerosol composition
would lead to decreasing SSA. This relationship is explored further in Section 3.2 and 3.3. Fig 2.
indicates that the light absorbed by methanol and acetone extracts were almost identical and would
imply that the amount and type of OC extracted by the two solvents were similar, as seen in other
studies as well (Chen and Bond, 2010; Wang et al., 2014). For some dung samples, the bias for
methanol and acetone extracts was close to 0.6 at SSA values of 0.95. These bias values were near
to the  theoretical prediction of 0.69 – 0.75 by Sun et al. (2007) for particle sizes much smaller
than the wavelength of light, even though our size distributions were not significantly smaller than
the wavelength of 405 nm. This could indicate that predictions by Sun et al. (2007) are valid for
sizes comparable to the wavelength of light as well, but more such observations are necessary to
obtain conclusive results. The reason for observed differences in the bias between water and
methanol extracts are discussed further in Section 3.3. The differences between the mean values
of $b_{abs,OA}/b_{abs,sol}$ at 375 and 405 nm were less than or close to the errors associated with them,
hence any trends with wavelength were not explored. In addition to this, there were no obvious
trends that could be explained using fuel type, leading us to not explore trends with fuel type either.
The value of $b_{abs,OA}/b_{abs,sol}$ approached a constant in the measured range of data. A power law
($y = k_0 + k_1 x^{k_2}$) was used to fit the points in Fig. 2, and the corresponding fit parameters, along
with root mean square error (RMSE) values, are listed in Table 1. The fit was performed using the
curve fitting tool in MATLAB and the RMSE values were calculated in Microsoft Excel. The
power law fits were deficient in capturing the true behavior of the bias with SSA but performed
better than corresponding mean values and step function curves. The parametrizations presented
in this section are representative of laboratory-based biomass burning (BB) aerosol emissions in
this study and are provided to mathematically visualize trends in the data. These parametrizations
might not be extensible to other emissions and should not be used for determining OA absorption
bias in other systems. The contribution of BC absorption coefficient to total absorption increases
with larger EC fraction of the aerosol which results in significant errors while extrapolating BC
absorption from longer wavelengths. Based on other studies, BC $A\mathring{A}E$ values range from 0.85 to
1.1 (Lack et al., 2008; Bergstrom et al., 2007; Lan et al., 2013). In Fig. 2, for data points below the
perforated lines at SSA values smaller than 0.7 at 375 nm and smaller than 0.825 at 405 nm, the
errors due to uncertainties is BC $A\mathring{A}E$ were greater than 30% and are a result of increasing BC
mass fractions at these SSA values. The large uncertainties at lower SSA values indicate that the
method described here is best suited to determine $b_{abs,OA}/b_{abs,sol}$ for particles with relatively
higher SSA values.

## 3.2  SSA parametrized with OC/TC

A linear relationship between aerosol SSA and EC/TC ratio was observed by Pokhrel et al. (2016).
To replicate the linear trends observed by Pokhrel et al., we studied the correlation between SSA
and OC/TC ratio (which is simply the EC/TC ratio subtracted from 1). Fig. 3 shows the variation
in SSA with change in the OC/TC ratio of the aerosol. The OC/TC ratio was determined using the
IMPROVE-A TOR protocol (Chow et al., 2007a) with a thermal optical EC/OC analyzer at Sunset
laboratories. The data was parametrized using an orthogonal distance regression (ODR) to account
for errors in the OC/TC ratio and resulting fits along with data points are plotted in Fig. 3. ODR is
different from a standard linear regression as it accounts for errors in both the independent and
dependent variables by minimizing least square errors perpendicular to the regression lines rather
than vertical errors as in standard linear regression. The ODR fits are linear with RMSE values of
0.04 and 0.02 for wavelengths 375 nm and 405 nm respectively. In Fig. 3, the points corresponding
to high OC/TC ratios are associated with SSA values that are close to 1, because pure OC aerosols
are predominantly light scattering. The fit yielded SSA values of 0.89 and 0.96 at 375 and 405 nm
respectively, for pure OA indicating that the fits represent a spectral dependence of absorption
which is characteristic of brown carbon optical properties because the SSA values for pure OC are
below 1 at both wavelengths and SSA at 375 nm is lower than that at 405 nm. (Chakrabarty et al.

315   2010).

A linear relation between the SSA and the EC/TC ratio (which is simply the OC/TC ratio
subtracted from 1) was also observed by Pokhrel et al. (2016). However, when the data from that
study were converted to OC/TC values for comparison, it was noted that the slopes and intercepts
of the resulting fits were different from those observed in this study. Table 2 has a list of the slope
and intercept of fits for comparable wavelengths in both studies, along with the RMSE for our fit.
A likely reason for dissimilar slopes and intercepts between the two studies could be due to
discrepancies in EC/OC ratios obtained using the same temperature protocol. Inter-comparison
studies have shown that different labs using the same sample with identical thermal protocols may
produce different results (Panteliadis et al., 2015). The instrument bias could be such that obtained
OC/TC ratios would have a proportional offset between different instruments leading to similar
linear trends but with different slopes which might be the case here. Another plausible reason for
the discrepancy could be positive artifacts in EC/OC analysis due to gas phase SVOCs being
adsorbed on the quartz surface because of phase partitioning of these compounds in the holding
tank. This reason seems less likely due relatively small sampling times for the aerosols. To assess
the performance of our parametrizations, we compared our fit to data obtained by Liu et al. (2014)
at 405 nm for BB aerosol. Data from the plots were extracted using Web Plot Digitizer (Rohatgi
2010) and was plotted with our fit in Fig. 4. We observed that our fits predicted SSA well at OC/TC
ratios > 0.7 with a RMSE value of 0.06 compared to 0.08 by Pokhrel et al. (2016) but predictions
were worse for 405 nm at lower OC/TC ratios as is also evident from the relatively high SSA value
of 0.39 for pure EC obtained using our parametrization. Most observations for soot SSA are lower
than those predicted by our 405 nm parametrizations (Bond et al., 2013, Schnaiter at al., 2003)
with our projections being closer to SSA observed by Radney et al. (2014). Generally, OC/TC
ratios are greater than 0.7 for laboratory and field BB (Xie et al., 2019; Akagi et al., 2011; Zhou et
al., 2017; Xie et al., 2017) which reduces concerns about underperformance of our fits for 405 nm
at low OC/TC ratios. It would be appropriate to use these parametrizations to determine a
reasonable range for SSA values rather than use them as a surrogate to determine actual SSA for
a given BB aerosol plume. A modification of Fig. 4 which compares the linear fits by Liu et al.
(2014) and Pokhrel et al. (2016) with our parametrizations is provided in the Supplementary
Information.
Despite the differences in ours and Pokhrel et al.'s (2016) fits, a useful conclusion from Fig. 3 is
that the OC/TC ratio determined using the IMPROVE-A protocol and SSA of BB aerosol have a
linear dependence. This dependence, however, has high variations at OC/TC ratios very close to
1, where fuel type and burn conditions dictate the composition and absorption properties (Chen
and Bond, 2010; Budisulistiorini et al., 2017) of organics released and hence a larger range of SSA
values exist at those OC/TC ratios. Further studies need to be conducted using more fuels with a
variety of distinct size distributions and burn conditions to determine the validity and exact
parameters for the fit.
**3.3  Absorption bias correlated with OC/TC ratio**
Fig. 5 depicts the variation in $b_{abs,OA}/b_{abs,sol}$ for primary OA with different OC/TC ratios. Given
the good correlation between OC/TC ratio and SSA, we expect to see a similar trend for Fig. 5 as
in Fig. 2.  As in Fig. 2, the bias in Fig. 5 increases with decreasing OC/TC ratio and approaches a
constant for the three solvents. A power law similar to the one in Fig. 2 was fit to the data in Fig.
5. The fit parameters for the different solvents at the two wavelengths, along with the RMSE values
corresponding to each fit, are presented in Table 3. We reiterate that the parametrizations for
$b_{abs,OA}/b_{abs,sol}$ as a function of OC/TC ratio depicted here are applicable to our system and should
not be used to calculate the bias in other systems. The exclusivity of depicted fit parameters to our
system excuses their relatively poor RMSE while representing the bias with OC/TC ratio. The
parametrizations are provided to represent some quantitative measure to the data rather than just
analyze the trends qualitatively. The large error bars from the Monte Carlo simulations at high EC
fractions are mainly due to uncertainties associated with the BC $A\mathring{A}E$. At lower OC/TC ratios, the
contribution of BC absorption to total particle-phase absorption coefficient is more pronounced,
leading to high uncertainties while extrapolating the coefficient to shorter wavelengths. It is
apparent from Fig. 5 that these errors in the bias are more prominent at OC/TC ratios below 0.75.
The burns with relatively high EC fractions are not representative of typical laboratory or field
BB. As mentioned earlier, typical laboratory BB have OC/TC ratios > 0.7 (Xie et al., 2017; Akagi
et al., 2011; Pokhrel et al., 2016; Xie et al., 2019) and > 0.9 for field BB (Aurell et al., 2015; Zhou
et al., 2017; Xie et al., 2017). Thus, data presented in Fig. 5 with relatively large errors and EC/TC
ratios > 0.25 are not representative of typical BB aerosol in either laboratory or field settings which
may warrant their exclusion from most analysis. We have still included these data points in our
plots and Tables but have excluded their use in data analysis due to the high errors associated with
them.
In Fig. 5, the difference in magnitude of the bias between methanol/acetone and water extracts
increase as EC fraction of the aerosol increases. An increase in the emissions of ELVOCs with
increasing EC/OC ratios was observed by Saleh et al. (2014) and we hypothesize that these
ELVOCs, which have high mass absorption efficiencies (Saleh et al., 2014; Di Lorenzo and Young
2016), could have a lower solubility in water than methanol or acetone which would explain the
increasing difference in $b_{abs,OA}/b_{abs,sol}$ values between water and methanol/acetone extracts.
Some of the generated ELVOCs might be insoluble in methanol and acetone as well, which would
lead to the observed increase in the OA absorption bias with decreasing OC fraction of the aerosol.
Based on the observed trends, these ELOCs would not be released indefinitely but tend towards a
constant above a given EC/OC fraction, mimicking an exponential behavior comparable to
observed trends in wavelength dependence for biomass burning OA with EC/OC ratios (Saleh et
al., 2014). This would lead to the bias approaching a constant value (due only to particle size
effects) with decreasing OC/TC ratios and in turn the aerosol SSA. Future studies can look at the
type and amount of ELVOCs released as a function of the EC/OC ratio of the aerosol and ascertain
if their solubility in these solvents is a function of their EC content.

## 3.4 Variations in AÅE with solvents and OC/TC ratios

The $A\mathring{A}E$ values for organics extracted in different solvents and those obtained from $b_{abs,OA}$ are
compared in Table 4. The $A\mathring{A}E$ values along with the errors for OA measurements were calculated
between λ = 375 and 405 nm using the Monte Carlo simulation. The $A\mathring{A}E$ for OC extracts were
calculated using Eq. 3 based on $b_{abs,sol}$ and corresponding errors were propagated based on
uncertainties in UV-Vis measurements. Consistent with previous studies (Chen and Bond, 2010;
Zhang et al., 2013; Liu et al., 2013), the $A\mathring{A}E$ values of water extracts were larger than the $A\mathring{A}E$ of
acetone and methanol extracts. Experiments by Zhang et al., (2013) observed that polycyclic
aromatic hydrocarbons (PAHs) absorbed light at longer wavelengths close to the visible region.
Organic solvents such as methanol have a higher extraction efficiency for these compounds than
water leading to higher absorption by methanol extracts at longer wavelengths which results in
lower $A\mathring{A}E$ (Zhang et al., 2013).
The $A\mathring{A}E$ calculated for OA ranged from 6.9 ± 1.7 to 15.6 ± 0.6 (excluding data with OC/TC >
0.75) which are slightly larger than $A\mathring{A}E$ values reported by most studies (Pokhrel et al., 2016;
Lewis et al., 2008). However, these studies report $A\mathring{A}E$ values in the visible range, which might
be lower than aerosol $A\mathring{A}E$ values in the UV range as observed by Chen and Bond (2010) for OA
extracts. The range of $A\mathring{A}E$ observed for water, acetone and methanol extracts were similar to those
observed by Chen and Bond (2010). A t-test for data presented in Table 4 shows that $A\mathring{A}E$ values
for OA were greater than their solution phase counterparts for both methanol (N = 17, $p$ = 0.0007)
and acetone (N = 17, $p$ = 0.0002). The difference in $A\mathring{A}E$ of OA and water extracts were statistically
insignificant (N = 17, $p$ = 0.25), but these differences were statistically significant at OC/TC ratios
≥ 0.9 (N = 12, $p$ < 0.05) where uncertainties due to BC absorption are lower. The reason for these
differences could be a combination of artifacts due to inefficient extraction of organics absorbing
light at lower wavelengths and the absence of size dependent absorption in the solvent phase which
might not capture effects of enhanced particle phase absorption at lower wavelengths. These bulk
solvent measurements of $A\mathring{A}E$ suggest that they might not be representative of spectral dependence
of OC in the particle phase, and future studies and models should be cautious while using $A\mathring{A}E$
data from solvent-phase measurements to be representative of the particle phase.
**3.5  Scaling factors based on Mie calculations**
To check the reproducibility of the conventionally used correction factor of 2, the absorption
coefficient determined from the bulk solvent absorbance using Eq. (1) was compared to absorption
coefficients calculated using Mie theory for three samples of smoldering sage. The EC/OC analysis
(IMPROVE-A protocol) determined that these samples consisted purely of OC, and because the
SMPS measurements and TOC analysis were only performed on the first set of samples, the three
samples of sage were considered optimum for Mie calculations.
The Mie based scaling factors for converting solution phase absorption coefficients to particulate
absorption for the three samples are presented in Table 5. TOC and EC/OC analysis indicated that
a similar fraction of organics at $61 \pm 2\%$ were extracted from all three samples. The Mie calculated
scaling factors at 375 nm and 405 nm are close to 2 as observed in previous studies (Liu et al.,
2013; Washenfelder et al., 2015) indicating that the conventional technique provides reproducible
results. The values for these scaling factor vary from 2 to 2.1 at 375 nm and 2.2 to 2.3 at 405 nm.
However, it is important to note that these scaling factors were not representative of actual biases
for determining OA absorption from solution phase as observed in Table 5. Thus, while a Mie
based correction factor of 2 can be duplicated, it is not representative of actual biases as also
corroborated by observations from Fig. 2 and Fig. 5. We recommend future studies to use caution
and judgement when using *a priori* scaling factors for determining OA absorption using solvent
extraction techniques.
**4 Conclusions**
Under controlled laboratory conditions, we determined artifacts associated with optical properties
of the solvent phase as compared to particle phase counterparts for fresh OA emissions from
biomass combustion. We combusted a range of different wildland fuels under different combustion
conditions, generating a span of different SSA and OC/TC values. The SSA values ranged from
0.55 to 0.87 at 375 nm, and from 0.69 to 0.95 at 405 nm, the OC/TC values ranged from 0.55 to
1. We observed an increasing difference in $b_{abs,OA}/b_{abs,sol}$ for water and methanol extracts with
increasing EC fraction of the aerosol. The decrease in absorption by water extracts with decreasing
OC/TC ratios was hypothesized to occur due to a decrease in extraction of ELVOC or similar
compounds with high mass absorption efficiencies by water. We also demonstrated that the SSA
and OC/TC ratios can be well parametrized with a linear fit that captures the effects of brown
carbon aerosol. We also determined that bulk solvent measurements of $A\mathring{A}E$ are not representative
of spectral dependence of OC in the particle phase. Finally, we analyzed the validity and
reproducibility of the conventionally used scaling factor of 2 for determining OA absorption
coefficients from water extracts of organics and noted that, while the factor is reproducible, its use
can misrepresent OA absorption coefficients. We recommend that future studies use caution while
applying a priori scaling factors to their systems as these factors might not be extensible to OA
emissions from all combustion processes. A comprehensive technique which improves extraction
efficiency with accurate knowledge of particle size distributions is necessary to determine correct
scaling relations.
For future experiments, a better technique to quantify BC absorption at lower wavelengths, such
as a thermodenuder to strip off all OC, or a single particle soot photometer along with core-shell
Mie calculations can be used to determine BC absorption and decrease uncertainties related to BC
absorption observed during experiments using this technique. Zhang et al. (2013) observed lower
$A\mathring{A}E$ for WSOC from a particle into liquid sampler (PILS) than for methanol extracts. The
hypothesis was that the highly dilute environment in PILS increased dissolution of organics in
water. This suggests that extraction of organics can be increased by heavily diluting the samples.
This can be combined with highly accurate spectrometers similar to the technique used by
Hecobian et al. (2010) to reduce some of the biases due to incomplete OA extraction.
**Data Availability**
All experimental data used to plot the figures in this manuscript (SSA, OC/TC ratios, and
$b_{abs,OA}/b_{abs,sol}$) are available for download at https://doi.org/10.17632/sdy3ptyrht.1 (Shetty

471  2019)

**Supplement**
The Supplement includes data on types of fuel combusted and their optical and physical properties
(Sect. 1, Table S1-S3), a pseudocode for the Monte Carlo simulations along with associated errors
in each variable (Sect. 2, Table S4), calculations for proxy size distributions used in Mie
calculations (Sect. 3), a modified version of Fig 4 depicting fits from referenced studies (Fig. S1)
and a graphical comparison of particle and solvent phase $A\mathring{A}E$ for the samples analysed in this
study (Fig. S2). The supplement related to this article is available online at
**Author Contributions**
RKC conceived of this study and designed the experiments. SB and WMH collected the fuels for
the experiments and performed EC/OC analysis on the sampled filters. NJS and AP carried out the
experiments. NJS analysed the data and prepared the manuscript with input from all co-authors.
**Competing Interests**
The authors declare that they have no conflict of interest
**Acknowledgements**
This work was partially supported by the National Science Foundation under Grant No.
AGS1455215, NASA ROSES under Grant No. NNX15AI66G.

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

**Figures and Tables:**

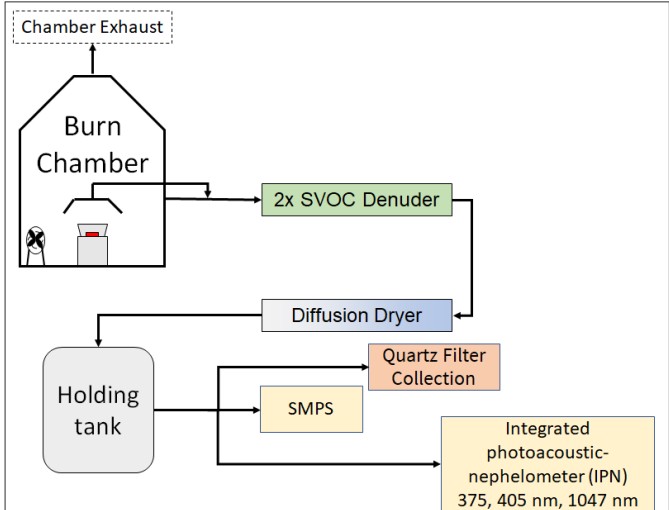


**Fig. 1:** A schematic representing the experimental setup. The aerosol emissions were either
sampled directly from the chamber wall or through a hood placed directly above the combusting
biomass.






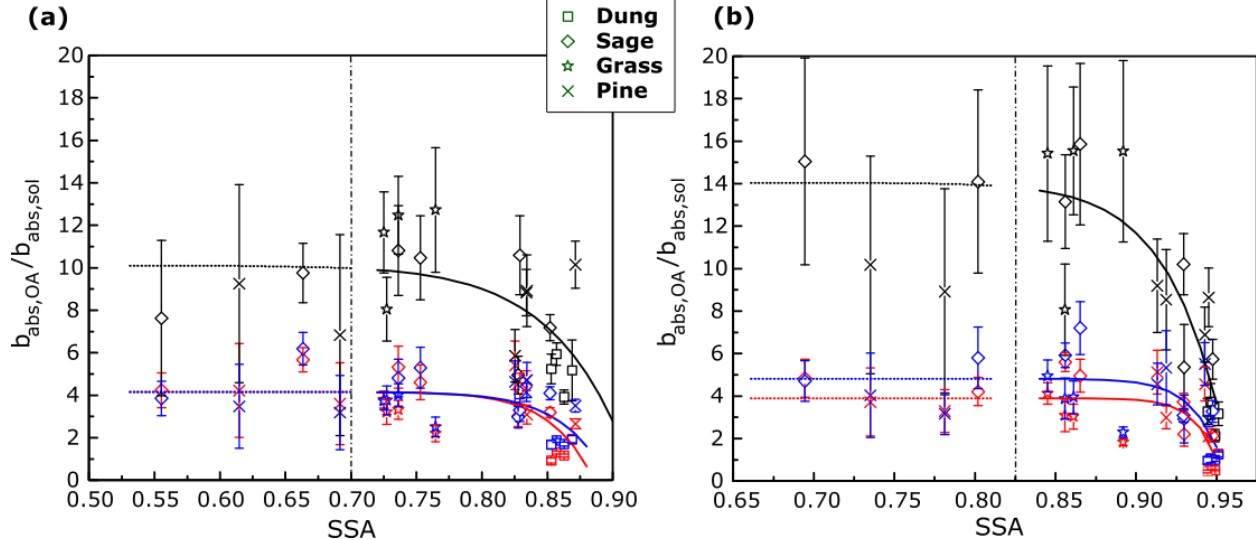

**Fig. 2**: Variation in $b_{abs,OA}/b_{abs,sol}$ with change in the SSA at (a) 375 nm and (b) 405 nm (N = 21). The error bars represent one standard deviation from the mean and were calculated using Monte Carlo simulations. The black markers represent water extracts, red markers represent acetone extracts and blue markers represent methanol extracts. The perforated lines separate points at lower SSA, which have high errors greater than 30% due to uncertainties in BC $A\mathring{A}E$, from the data at high SSA.

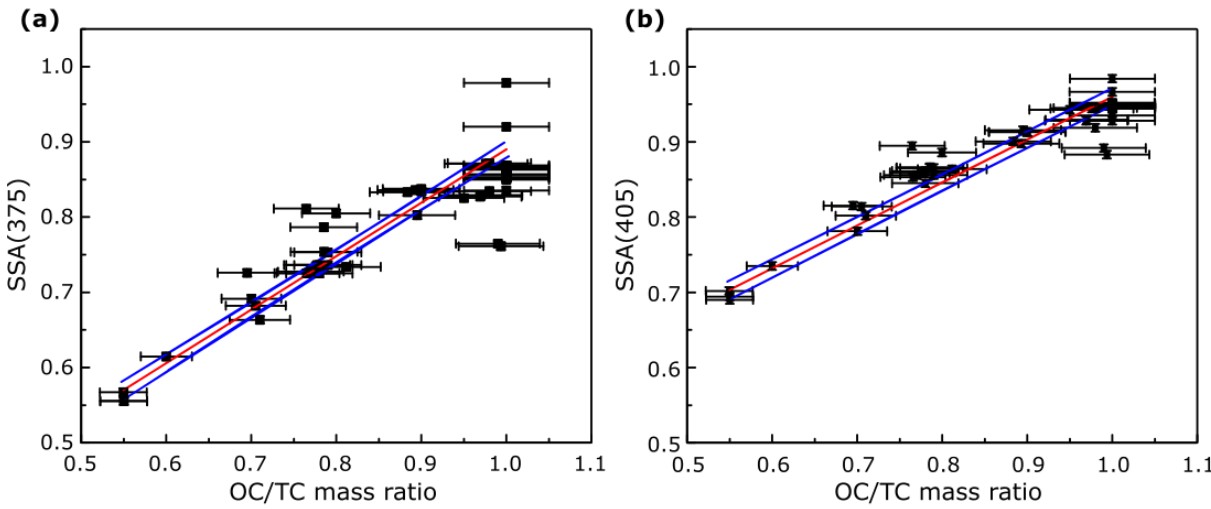


**Fig. 3:** SSA at (a) 375 nm and (b) 405 nm as a function of the OC/TC ratio (N= 49). The solid red
lines are ODR fits to the data and the solid blue lines represent the 95% confidence intervals. The
errors in OC/TC ratios were determined by the quadrature sum of uncertainties from EC/OC
analysis and the error in SSA were negligible

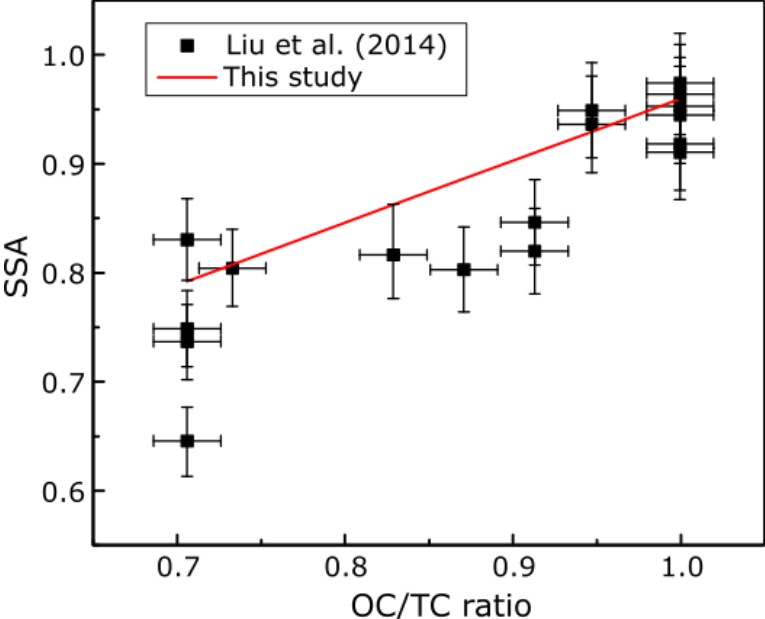


**Fig. 4:** Measured SSA values by Liu et al. (2014) for controlled laboratory combustion
experiments (black squares) and overlaid with the points is the ODR parametrization determined
in this study (solid red line).

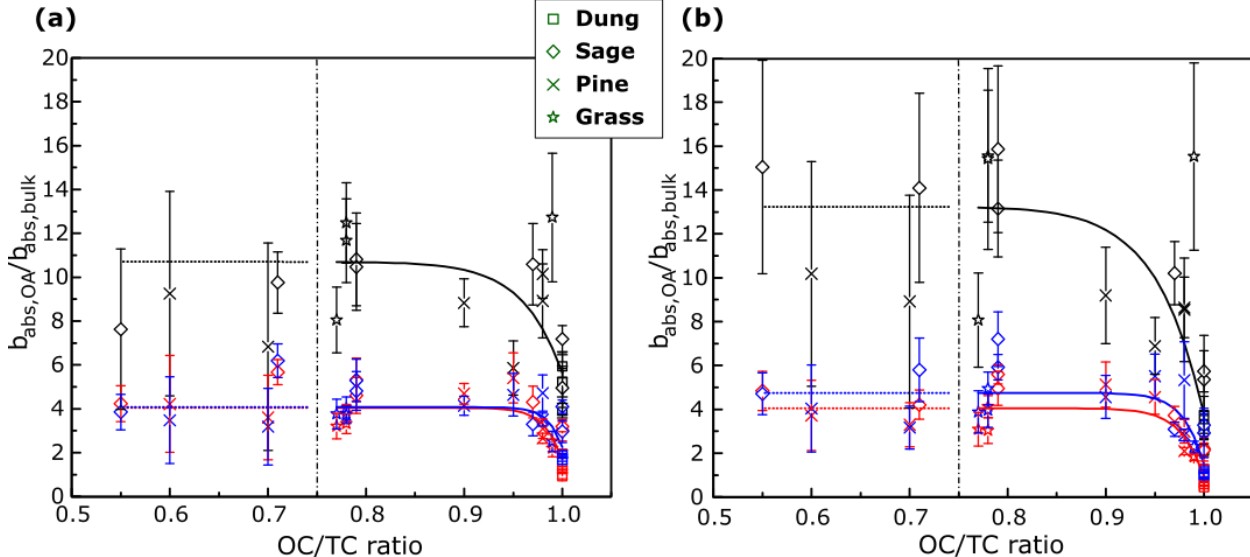

**Fig. 5:** The values of $b_{abs,OA}/b_{abs,sol}$ plotted with the OC/TC ratio, instead of the SSA, as in Fig. 2. Black markers represent data for water extracts, red markers represent data for acetone extracts, and blue markers represent data for methanol extracts.

Table 1: Fit coefficients for $b_{abs,OA}/b_{abs,sol}$ as a function of SSA ($y = k0 + k1\,(SSA)^{k_2}$) for
tested solvents and the fuels analyzed in this study along with the RMSE value for each fit.

|  | Wavelength (nm) | Solvent | Fit Parameters | | | |
| --- | --- | --- | --- | --- | --- | --- |
|  |  |  | $k_0$ | $k_1$ | $k_2$ | RMSE |
| $\dfrac{b_{abs,OA}}{b_{abs,sol}}$ | 375 | Water | 10.1 (±2.1) | -39.8 (±177.1) | 16.1 (±31.3) | 2.2 |
|  |  | Acetone | 4.2 (±0.8) | -117.4 (±36.9) | 27.5 (±37.5) | 1.1 |
|  |  | Methanol | 4.2 (±0.8) | -69.1 (±451.6) | 25.8 (±45.4) | 1.1 |
|  | 405 | Water | 14 (±4.2) | -42.4 (±70.5) | 27.4 (±35.5) | 2.6 |
|  |  | Acetone | 3.9 (±1.1) | -95.6 (±609.9) | 68.3 (±121.8) | 1.3 |
|  |  | Methanol | 4.8 (±1.4) | -49.1 (±250.3) | 53.1 (±98) | 1.5 |


Table 2. ODR regression coefficients along with errors in brackets for plots of SSA v/s OC/TC
ratios (y = m (OC/TC) + c) for the different biomass fuels used in this study, and parameters for
ODR fit from Pokhrel et al. (2016) for 405 nm, along with RMSE values for our fits.

|  | Wavelength (nm) | m | c | RMSE |
|---|---|---|---|---|
| This study | 375 | 0.71 (±0.04) | 0.18 (±0.03) | 0.04 |
|  | 405 | 0.57 (± 0.02) | 0.39 (± 0.02) | 0.02 |
| Pokhrel | 405 | 1.07 (±0.04) | -0.13 (±0.04) | ~~ |


Table 3: Fit parameters for ratios of the absorption coefficient of organics in the particle phase to
the absorption coefficient of the solvent phase, as a function of the OC/TC ratio ($y = k_0 +$
$k_1(OC/TC)^{k_2}$) for the fuels analyzed in this study, along with the RMSE value for each fit.

| | Wavelength (nm) | Solvent | Fit Parameters | | | |
| --- | --- | --- | --- | --- | --- | --- |
| | | | $k_0$ | $k_1$ | $k_2$ | RMSE |
| $\dfrac{b_{abs,OA}}{b_{abs,bulk}}$ | 375 | Water | 10.7 (±1.8) | -5.1 (±2.5) | 25 (±36) | 2.0 |
| | | Acetone | 4.1 (±0.8) | -2.4 (±1.2) | 58 (±86.2) | 1.0 |
| | | Methanol | 4.1 (±0.7) | -1.8 (±1.1) | 71.5 (±139.6) | 0.9 |
| | 405 | Water | 13.2 (±2.4) | -9.5 (±3.2) | 20.8 (±20.9) | 2.5 |
| | | Acetone | 4.1 (±0.9) | -3.1 (±1.3) | 43.3 (±49.6) | 1.0 |
| | | Methanol | 4.8 (±1.1) | -3.2 (±1.7) | 49 (±69.5) | 1.3 |


Table 4: The $A\mathring{A}E$ of OA from various fuels extracted in water, acetone, and methanol, along with
the $A\mathring{A}E$ calculated for $b_{abs,OA}$.

| Fuel | OC/TC ratio | $A\mathring{A}E_{375-405}$ | | | |
| --- | --- | --- | --- | --- | --- |
| | | OA | Water | Acetone | Methanol |
| Dung | 1 | 13.7 ± 2.3 | 8.0 ± 2.0 | 5.3 ± 1.4 | 5.2 ± 1.3 |
| | 1 | 15.3 ± 2.4 | 9.0 ± 2.0 | 5.9 ± 0.4 | 7.8 ± 0.6 |
| | 1 | 15.6 ± 0.6 | 7.5 ± 1.8 | 4.6 ± 0.3 | 4.5 ± 0.9 |
| | 1 | 14.9 ± 2.7 | 8.6 ± 1.2 | 5.3 ± 0.2 | 6.8 ± 0.4 |
| Sage | 1 | 13.9 ± 1.9 | 10.9 ± 1.2 | 8.6 ± 0.7 | 8.8 ± 1.1 |
| | 1 | 10.7 ± 1.5 | 10.7 ± 4.5 | 6.3 ± 3.2 | 7.3 ± 2.9 |
| | 0.97 | 10.6 ± 2.4 | 9.9 ± 1.4 | 5.2 ± 0.8 | 5.8 ± 0.7 |
| | 0.79 | 7.4 ± 2.9 | 12.3 ± 2.4 | 8.6 ± 0.8 | 9.2 ± 1.2 |
| | 0.79 | 8.2 ± 2.4 | 10.6 ± 2.2 | 8.7 ± 0.8 | 8.3 ± 1.3 |
| | 0.71 | 10.4 ± 1.4 | 7.5 ± 3.1 | 6.3 ± 1.7 | 6.4 ± 2.1 |
| | 0.55 | 9.9 ± 4.2 | 6.5 ± 4.8 | 3.8 ± 2.0 | 3.6 ± 2.8 |
| Grass | 0.99 | 10.1 ± 2.4 | 12.1 ± 4.6 | 7.8 ± 0.9 | 7.5 ± 1.2 |
| | 0.78 | 9.9 ± 3.1 | 10.2 ± 2.3 | 8.5 ± 0.5 | 9.6 ± 0.6 |
| | 0.78 | 6.9 ± 1.7 | 9.7 ± 3.8 | 7.5 ± 0.8 | 7.3 ± 1.6 |
| | 0.77 | 9.0 ± 4.0 | 8.2 ± 1.6 | 8.1 ± 0.6 | 8.4 ± 0.9 |
| Pine | 0.98 | 11.8 ± 1.0 | 9.4 ± 2.0 | 8.6 ± 0.8 | 8.1 ± 1.1 |
| | 0.98 | 8.7 ± 1.9 | 9.6 ± 3.4 | 8.4 ± 1.8 | 8.6 ± 1.5 |
| | 0.95 | 14.2 ± 3.5 | 16.4 ± 1.3 | 11.8 ± 0.9 | 12.8 ± 1.3 |
| | 0.9 | 8.2 ± 2.4 | 9.1 ± 2.3 | 8.8 ± 1.6 | 8.7 ± 2.0 |
| | 0.7 | 16 ± 10.9 | 9.9 ± 3.1 | 6.3 ± 2.2 | 5.8 ± 2.1 |
| | 0.6 | 17.4 ± 10.8 | 6.4 ± 3.3 | 5.2 ± 2.8 | 5.4 ± 2.9 |




Table 5: Correction factors for bulk solution absorption to particle phase absorption, based on Mie
Theory calculations.

| Fuel | Geometric mean (in nm) | Geometric standard deviation | Mie based Scaling Factor | | IPN based bias | |
|---|---|---|---|---|---|---|
| | | | 375 nm | 405 nm | 375 nm | 405 nm |
| Sage | 397 | 1.3 | $2.0 \pm 0.4$ | $2.3 \pm 0.4$ | $2.6 \pm 0.6$ | $1.6 \pm 0.6$ |
| | 271 | 1.32 | $2.1 \pm 0.4$ | $2.3 \pm 0.4$ | $2.8 \pm 0.6$ | $1.9 \pm 0.3$ |
| | 159 | 1.59 | $2.0 \pm 0.4$ | $2.2 \pm 0.4$ | $2.8 \pm 0.5$ | $1.8 \pm 0.4$ |



