# Peer review of "Measuring Light Absorption by Freshly Emitted Organic Aerosols: Optical Artifacts in"

_Atmospheric Chemistry and Physics, 2018_

## Referee Comment (RC1) · Anonymous Referee #1 · 21 Dec 2018

While I appreciate that the authors have looked into differences between suspended and solution-phase absorption by OA, and have been waiting for someone to do this in more detail, I am ultimately not in support of the publication of this paper in its current form. For one, I am concerned about the suggestions for such a wide range of scaling factors for water extracts. This combines too many factors, specifically size and solubility. If not all of the material is soluble, then the correction factors are forcing agreement for something that was not even measured, that is the amount of material extracted. While this might have worked for these particular test samples, I have serious doubts about the robustness of the scaling factors. They talk about the need for scaling factors as a function of SSA. But, as these sampled particles are aged in the atmosphere,

or mix with other sources should I expect the relationships determined here to hold? I am skeptical. After reading through a few times, I've actually become increasingly skeptical. Or at least the presentation of the results. In my view, this study points out that one should take caution when using water or methanol extracts to determine absorption. Most likely it will lead to a low bias compared to the atmosphere. While we already knew this, this study adds by showing the issue more explicitly than many previous studies. But I don't actually think we should be presenting "correction factors" that might be used because I think that these are likely to lack robustness, especially in the case of water as a solvent. Ultimately, I unfortunately do not think that this paper should be published in it's current form. My concern is that the casual reader might just apply these scaling factors to their data without understanding that they very will might not apply at all due to differences in their system. I suggest that the authors refocus the paper, drawing out the distinction between a bias (which results from extraction) versus the difference between solvent and particles as an explicitly optical issue (i.e. the need for Mie theory or something similar). Additionally, the authors need to provide a more robust uncertainty analysis.

In Fig. 2a and 4a (which are related), the data to me seem to just show a step function for the water extracts. There are a subset of measurements where the OA/bulk are relatively low and then a step up to a bunch of measurements with higher ratios. This does not look like a power law to me at all. Also, what should I make of the measurements where OA/bulk = 1? The authors have made a case that the particle/solution difference should give a minimum difference of a factor of 2 (if extraction were 100%).

L47: The authors should decide whehther this statement is in reference to observational studies or model studies and cite accordingly. They mix observation with model here, making it less clear what their point is. If observational, they should cite the now famous Kirchstetter paper.

L52: The authors might consider citing the work from the Heald group and from Saleh. They will find the list of studies that intentionally include absorbing OA is rapidly increasing.

L56: Suggest changing to "ideally excludes…EC." It is possible that EC can break through filters and impact measurements. See the work by Geoff Smith (Phillips and Smith, AS&T, 2017).

L57: I strongly suggest modifying this statement about this being a "good" analytical method to state right up front that it does not measure "only the OC absorption spectra." It measures the absorption spectra of the OA that is extracted into the solvent. If the method were "good" and measured "only the OC absorption spectra" then there would be no dependence on solvent (or pH).

L62: One must also assume something about the real component. Suggest changing to "complex refractive index." Also, why an "assumed" number distribution. Why not a measured one?

L64: In mentioning "past studies," it is not clear whether the authors here are referring to some issue with the samples not being suspended particles or to issues associated with extraction of only a subset of the total OA material. The former is a method limitation. The latter is a bias. These should be distinguished here. Related, in the next sentence the authors mention past studies obtained different correction factors for water and methanol. But if Mie theory were the only issue then there would be no difference. This again emphasizes that the issue of incomplete extraction must be brought up and made a central part of this discussion.

L78: It is not clear to me that either the Zhang paper or Saleh paper address the issue of "the types and fractions of organics extracted by a given solvent" and how these relate to SSA or EC/OC. They address variability in properties, yes. But I don't think they address what the authors purport.

L84: It is not necessarily correct to state that a single-wavelength PAS cannot separate OC from BC absorption. One can, at least in theory, evaporate OC to just determine the

[Figure]

BC absorption. Alternatively, if one can make high quality measurements of the MAC at a single wavelength, then this can be compared to an appropriate reference value. These are both as valid as extrapolations from multiple-wavelength measurements. (All must be interpreted with caution and attempt to account for coating effects.) I find this sentence and the ones that follow to be overstating the case and pushing a particular view of how things should be measured, but stating it as an objective fact. This should be revised.

L142: Was a sonicator used? It would be surprising to find out that the samples were not sonicated during extraction.

The authors should report the extraction efficiency for the water solvent, as they can do this from the WSOC and OC measurements. What fraction of OA was extracted? It presumably must be small, or the correction factor for water versus methanol would not be all that large.

L178: "OC/TC ratios were assumed constant...". Was this assumption tested in any way?

L198: For the blacker samples (lower OC/TC), the BC will absolutely impact the retrieval of the real refractive index. How was this accounted for?

L212: The Cheng et al. reference is to a computational study. As much as those can shed insights, I suggest using an observational or lab study to make the case of the value for the SSA for BC. With the exception of some recent results from NIST (Radney et al., 2014, ES&T), I think that most experimental studies suggest lower values than stated here are possible.

L222: Presumably this power law was arbitrarily chosen? Were other forms explored? What is the predictive power of this, especially at high OC/TC? See the above comment about the data seeming to look more like a step function, than a power law.

**L231: These are not "fluctuations." They are simply uncertainties. However, the error

bars reported do not seem to reflect these uncertainties properly. There is no notable decrease in the size of the error bars below/above the thresholds identified. This begs the question, how were the uncertainties determined? The currently reported uncertainties in Fig. 2 are clearly underestimated, based on the potential for a 200% bias. Ultimately, the uncertainty is likely a direct function of the OC/TC, since the BC contribution will be larger when this ratio is smaller, and thus it will become increasingly difficult to separate OC from BC contributions. More than that, any uncertainty in the AAE will create a systematic, but OC/TC-dependent, bias in the key ratio determined here. I think that these issues need to be discussed in much greater detail.

L242: This statement by the authors, that if they translate data from another study to the parameter space used here (OC/TC) they find different results, suggests that the premise of this study might be flawed. This suggests that the results here might not translate well to other settings, and thus the fit function determined in Fig. 2 is not robust beyond the current study. The authors need to address the issue of how robust they expect their parameterization to be, and how extensible to other systems. Also, the fact that the current study and a previous study disagree so much seems to limit the statement on L253 that the SSA can be predicted from the OC/TC. The prediction from the fits in this study may simply not be robust.

L279: The authors need to clarify how specifically an OC/TC dependence of the OA absorptivity explains the apparent difference in methanol versus water.

L284: Are the authors deriving their conclusion that the Zhang et al. (2013) results support the findings here from the following sentence in Zhang: "The water-insoluble BrC, calculated as the difference between methanol- and water-extracted BrC, exhibited a tighter correlation with ambient EC concentrations ($r2 = 0.81$, Figure 5b) than water-soluble BrC ($r2 = 0.40$), suggesting that the water-insoluble BrC components and EC have similar sources (e.g., incomplete combustion from vehicle emissions and wood burning)." As best I can tell, this is the only sentence that might connect. But I am skeptical of the relevance, since in the Zhang case the distinction is largely between primary and secondary OA, not different types of primary OA. I suggest that the authors' argument needs to be strengthened if it is to be kept.

L294: It is not clear how the authors come to the conclusion that "higher molecular weight" compounds are responsible here. This seems like speculation and should be posed as such.

As a general comment, throughout I suggest that the authors get rid of the "bulk" language and refer to these as liquid-extracted samples. Or "solution phase." Or something else similarly descriptive. "Bulk" is not especially descriptive, since particles also contain "bulk" material.

L288 and AAE discussion: The authors do not present an error analysis here, and the uncertainties on the lower OC/TC samples will be very large if propagated appropriately. This is especially important for any of the conclusions reached regarding comparison between particle and solution-phase differences.

L300: The authors state that the particle phase AAE are "close to" those in the solution phase at high OC/TC. But then in the next sentence they state that the particle phase and solution-phase "deviate significantly." These seem contradictory. I actually do like this general aspect of the analysis (especially at higher OC/TC, where uncertainties from extrapolation are smaller). But, without a robust error analysis and a more quantitative discussion of the comparison I don't think the authors can arrive at their conclusions. At minimum, there should be something like a t-test to check for statistical differences between the particle- and solution-phase AAE values. I suspect that if the authors include all data (not just dung) they will find that the methods give statistically indistinguishable AAE values.

The Mie theory section is fine, but I think that to some extent it shows the overall inadequacy of this paper. If the issue is that e.g. water extraction does not extract all absorbing organic species, then a correction factor is not what is necessary, better extraction is necessary. The idea that there might be some universal correction factor

that can account for solubility differences among the many different sources and types of OA in the atmosphere is, in my view, not realistic.

---

## Referee Comment (RC2) · Anonymous Referee #2 · 22 Dec 2018

Shetty et al investigated the correction factor for converting from bulk to particle phase absorption coefficients for primary OA emitted from biomass burning. Aerosol samples were generated experimentally by using the combustion chamber, as explained in Sumlin et al. 2018b, also uses a variety of aerosol sources such as sage, cattle dug, and grass. The study was concentrated for the correction factor as a function of single scattering albedo (SSA), where SSA values are obtained from filter based observations by using integrated photoacoustic-nephelometer spectrometers (IPNs). The manuscript possesses scientifically relevant data based on experimental observation and the Mie theory. The topic of the paper is scientifically important. Scientific data and data products that are presented in the manuscript lead to a fair rating of the

manuscript. The manuscript is well organized. The abstract and conclusion provide a reasonably complete summary of findings. Figures are clear, however, there are some missing scientific explanations of observation techniques, data analysis and uncertainty of the data in the manuscript. In summary, this manuscript, after some major revisions (as outlined below and comments posted from Anonymous Referee# 1, for this manuscript) fulfills the quality requirements for publication in ACP.

Specific Comments:

*Lines 114/116: The ratio of OC/TC were reported of ranges 0.55-1 but figure 3 (line 489) shows that ratios observed are only in the specific data ranges such as 0.55, 0.6, and 0.7. 0.8, and 1. It is not clear why the OC/TC data of ratios in between those ranges, such as of ratios 0.65, 0.75, and 0.85, were not observed and not reported. Are these ratios were rounded? And also, there is no excess data of ratios about 0.55 and 0.6 which contribute for analysis. I have the impression that the significant correlation, ∼0.95 of SSA vs OC/TC is mainly driven by some outliers. I strongly recommend presenting error analysis on these data sets.

*Lines 124/126: Why was SMPS not used for all of the experiment? Please clarify this. Also, give a reference or a brief explanation of how the geometric mean size distribution was determined?

*Line 151: There are no references or derivation of mathematical equations used in the manuscript, for example, Eqn.1, babs, bulk $(\lambda)$. Please provide the references or derivations to support the validity of the mathematical equations used in the manuscript.

*Line 154: Explain why absorbance at a given wavelength is normalized to the absorbance at 700nm. *Lines 183/185: What is the range of assumptions made along with Mie theory, as stated in the text? Reference is recommended to include for determining the imaginary complex refractive index.

[Figure]

*Line 223: How were the RMSE values calculated? Please include the reference/formula or name of software which was used to get RMSE values in Table 1, such as excel MATLAB, or Igor Pro.

*Lines 223/224: Please add a line to justify the impact of BC AAE on conversion factor for particles with SSA smaller than 0.7 at 375nm and smaller than 0.825 at 405 nm.

*Line 225: What is sensitivity analysis as stated in the text? Please explain briefly.

*Line241/242: Briefly describe why the two different studies, Pokhrel et al. (2016) and the current studies give different slopes and intercepts of the resulting fits?

*Line 276: What does SI represent for?

*Lines 292/295: Briefly explain why the higher molecular weight compounds absorb more light?

*Lines 296: AAE values for OA are significantly high with wide ranges of 4.4 -14.61. How are these values related to wavelengths? Please provide some references, if there are any, to support these values.

*Lines 296/297: It is reported that overall AAE for OA decreases with increased EC. Please add a graph/or a brief note to show AAE for OA measurements with the concentration of EC.

*Line 342: Authors' names are not clearly reported: RKC, SB, WMH, NS, AP, are not previously reported with these names in the authors' list. I think it is not relevant to include author contributions in the manuscript once a list of authors is reported.
* * *

---

## Referee Comment (RC3) · Anonymous Referee #3 · 24 Dec 2018

Anonymous Referee #1 has already submitted a review that articulates very clearly the overall discomforts I have with this manuscript. To avoid repeating that excellent discussion, I will simply propose this alternative title for the work:

Measuring Light Absorption by Organic Aerosols: Observed Total Error in an Illustrative Technique Using Extraction-Based Photometry

I am not sufficiently familiar with the literature of extraction-based absorption photometry to judge whether the accuracy of similar techniques is widely over-rated. If that is the case, then a revised version of this discussion paper could provide the convincing and well-documented critique that the field then needs. Otherwise, I would not recommend

publication.

Specific Comments

(1) The authors effectively define the in-situ shortwave absorption coefficient for organic aerosol, babs,OA, as the excess of the PAS-measured total over the BC contribution. They extrapolate the necessary shortwave BC value from a longwave PAS measurement, via an assumed unit AAE. This BC contribution is the proverbial 'elephant in the room', appearing nowhere in the results but providing essential context for their interpretation. What are the relative contributions of OA to total absorption at short wavelengths, and how do they vary with fuel type and burn conditions? Extended AAEs (AAE405-1047 or AAE375-1047), from which a curious reader could derive an answer, are nowhere indicated. A related question is how longwave absorption and TOR EC relate to each other, since they are independent proxies for the same BC. Figure 3 plots only ratios against ratios, shortwave bscat/(bscat+babs) from the in-situ IPN measurement against OC/TC from the filter TOR analysis. It would seem at least equally instructive to compare the concentration values directly, longwave babs against TOR EC.

(2) Two SVOC denuders sit between the burn chamber and the holding tank (Figure 1), but we can expect some phase re-equilibration to occur within the holding tank before samples are drawn. Adsorption of re-volatilized organic species by the quartz sampling filters will then generate artifacts in the TOR and extraction measurements that are not present in the IPN optics measurements. Did the authors collect and analyze quartz blanks to quantify these artifacts, using a (non-adsorbing) PTFE filter between the smoke-filled holding tank and sampling port to exclude the particle phase?

(3) It is hard to relate and reconcile the experimental data shown or listed in the different figures and tables. Not all IPN- and TOR-characterized burn samples were filter-extracted for OA, and not all water extracts were analyzed for TOC. It would be helpful to make these experimental layers clearer to the reader, along with some indication of

criteria for inclusion/exclusion. For example, Table S1 lists 53 filters collected from 28 burns. I infer that this (53) counts just the filters consumed for TOR OC/TC analysis, each paired with another filter collected for extraction (lines 133-134). (If the 7 filters from dung burns instead represented all TOR and extraction filters together, then we would have at most 3 pairs yielding complete records, contra the 4 observations listed in Table 4.) It appears from Table 4 that only 21/53 $\sim$ 40% of the other filters were selected for quartering and extraction. Does Figure 3 show all 53 observations from Table S1? Do Figures 2 and 4 show those 21 observations from Table 4? It would be helpful to be told the total number of observations appearing in each figure and table.

(4) Is it true that the TOR analyses were performed AT Sunset Laboratories (line 238), and not locally with an instrument manufactured BY Sunset Laboratories?

(5) In addition to flowrate and sampling time (line 133), filter area is a relevant experimental factor and should be specified.

---

## Referee Comment (RC4) · Anonymous Referee #4 · 3 Jan 2019

Given the comments already made by the other three anonymous reviewers, I will refrain from repeating what they have stated. I agree with Reviewer #1's assessment that the conclusions from this work are not sufficiently general to be of use for correcting bulk, solvent-based absorption measurements. As the other reviewers have pointed out, the measured correction factors incorporate not just geometric differences in bulk and particle absorption but also solvent- and constituent-specific factors, including solubility. And, there are correction factors measured at nearly identical SSA or OC/TC values that differ by factors of 2-3 (Figures 2 and 4) – such scatter is too great to draw a meaningful conclusion about the dependence of the correction factors on SSA or OC/TC ratio. It appears as if no dependence, i.e. a horizontal line, would describe the

id="2" /
trends about as well as the arbitrarily-chosen power law function.

In short, the main conclusion from this study is that there are different correction factors for water and methanol/acetone with water extracting less absorption than the other solvents. This conclusion is not new and may not be general to other types of absorbing organic aerosols or even other types of biomass burning aerosols. What is more, the extent of scatter makes potential use of these factors problematic. Hence, the factors measured here are not broadly applicable. Furthermore, the purported dependence of these factors on SSA or OC/TC is overstated making that conclusion suspect as well.

————————————————————

---

## Author Comment (AC1) · 30 Mar 2019

We thank the reviewers for their extremely insightful suggestions. Based on the reviewers' comments we have made substantial changes to the manuscript including modifications to the data analysis and conclusions. Briefly, we employed a Monte-Carlo approach to estimate the uncertainties in the ratio of the organic aerosol absorption in solution phase to that in aerosol phase, due to (1) measurement errors and (2) uncertainty in the true (and unknown) value of BC AÅE. Instead of suggesting that these parameterizations are universally-applicable correction factors, we now only use them to explore the relationship between solvent method artifacts and intrinsic properties of the aerosol samples. Many studies which are now referenced in the updated manuscript have used such "correction factors" without regard to artifacts associated with individual system biases. We believe this manuscript provides the necessary critique for the use of such factors and elaborates upon AÅE based artifacts in solvent extract measurements which have not been actively studied in past work in this field. The title, abstract and conclusions have also been modified to reflect these changes.

Below are our responses to specific concerns brought up by the reviewers. The comments by reviewers are in boldface followed by authors' responses:

**Reviewer 1:**

**In Fig. 2a and 4a (which are related), the data to me seem to just show a step function for the water extracts. There are a subset of measurements where the OA/bulk are relatively low and then a step up to a bunch of measurements with higher ratios. This does not look like a power law to me at all. Also, what should I make of the measurements where OA/bulk = 1? The authors have made a case that the particle/solution difference should give a minimum difference of a factor of 2 (if extraction were 100%).**

We had considered the use of a step function over a power law, the reason for selecting the power law fit was to keep the equation consistent through all regressions. It would appear that a step function is better than the power law for the water extracts, but the gradual slope with OC/TC ratios and SSA is more prominent in the methanol and acetone extracts. In addition to this, the root mean square error (RMSE) values for the power law fit were consistently lower than their step function counterparts. The power law fit can also mimic a step function with a steeper slope and can also have a curve with a gradual slope depending on the value of the power law exponent. None of our OA/bulk values for water were below 2, hence we had suggested a range from 2 to 11 for water extracts, and 1 to 4 for methanol/acetone extracts. We realize now that reporting these as correction factors without measurements for extraction efficiency and particle size distribution would be misleading and have removed these suggestions all together.

**L47: The authors should decide whether this statement is in reference to observational studies or model studies and cite accordingly. They mix observation with model here, making it less clear what their point is. If observational, they should cite the now famous Kirchstetter paper.**

The citation has been added and all citations are now referencing observation studies

**L52: The authors might consider citing the work from the Heald group and from Saleh. They will find the list of studies that intentionally include absorbing OA is rapidly increasing.**

Citations have been added and the sentence modified from "have a few global modeling studies…" to "have global modeling…"

**L56: Suggest changing to "ideally excludes. . .EC." It is possible that EC can break through filters and impact measurements. See the work by Geoff Smith (Phillips and Smith, AS&T, 2017).**

The sentence has been changed. The authors were familiar with the AS&T paper by Phillips and Smith, and avoided ultrasonication of the filters to prevent mechanical dislodging of deposited BC aggregates

**L57: I strongly suggest modifying this statement about this being a "good" analytical method to state right up front that it does not measure "only the OC absorption spectra." It measures the absorption spectra of the OA that is extracted into the solvent. If the method were "good" and measured "only the OC absorption spectra" then there would be no dependence on solvent (or pH).**

The original intent of using different solvents in our experiments was to quantify the difference in biases for this method due to varying extraction efficiencies of different solvents. Perhaps we were not clear in conveying the intent for the use of different solvents in the previous iteration of the manuscript. We have modified the entire paragraph to distinguish between extraction related biases and limitations due to size measurement errors.

**L62: One must also assume something about the real component. Suggest changing to "complex refractive index." Also, why an "assumed" number distribution. Why not a measured one?**

The sentence has been changed to reflect that an assumed real part as well as the measured imaginary part of the refractive index are used together to calculate the Mie based absorption coefficient. An "assumed" number distribution was written to justify that the extracted organics could/would have a distinct size distribution than the organic aerosol (OA). Scientists can use an SMPS based size distribution or a PM based size distribution for OA, each with their own assumptions. We have however, changed the sentence to reflect that the distributions can be measured or assumed.

**L64: In mentioning "past studies," it is not clear whether the authors here are referring to some issue with the samples not being suspended particles or to issues associated with extraction of only a subset of the total OA material. The former is a method limitation. The latter is a bias. These should be distinguished here. Related, in the next sentence the authors mention past studies obtained different correction factors for water and methanol. But if Mie theory were the only issue then there would be no difference. This again emphasizes that the issue of incomplete extraction must be brought up and made a central part of this discussion.**

We agree with the reviewer completely and thank them for pointing out the insufficiency in our arguments for this case. We have restructured and reconstructed the entire paragraph to emphasize the reason for these biases. Citations have been modified to reflect differences both due to incomplete OC extraction as well as size-dependent absorption properties.

**L78: It is not clear to me that either the Zhang paper or Saleh paper address the issue of "the types and fractions of organics extracted by a given solvent" and how these relate to SSA or EC/OC. They address variability in properties, yes. But I don't think they address what the authors purport.**

The authors thank the reviewer for pointing this out. We have modified the sentence to "…such as the EC/OC ratios, and single scattering albedo (SSA), even though these properties have shown to be well correlated with OA optical properties." to signify that the EC/OC ratios are correlated with the "brownness" of organics with production of more ELVOCs, and these in turn could impact the amount of OC extracted by traditional solvents. The meaning of the sentence seems to have altered over subsequent iterations to the manuscript hence the mismatch in citations.

**L84: It is not necessarily correct to state that a single-wavelength PAS cannot separate OC from BC absorption. One can, at least in theory, evaporate OC to just determine the BC absorption. Alternatively, if one can make high quality measurements of the MAC at a single wavelength, then this can be compared to an appropriate reference value. These are both as valid as extrapolations from multiple-wavelength measurements. (All must be interpreted with caution and attempt to account for coating effects.) I find this sentence and the ones that follow to be overstating the case and pushing a particular view of how things should be measured, but stating it as an objective fact. This should be revised.**

We have changed the sentence to state that a single-wavelength PAS on its own cannot separate OC and BC absorption. In the Conclusion section of the earlier manuscript, the technique mentioned by the reviewer and a few more were stated as potential alternatives for measuring BC absorption. We have conditioned the language of the paper to reflect that this is one of the potential methods with which BC and OC absorption can be separated. The authors are confident with this method as it is free of biases related to thermophoretic particle losses as seen with most thermodenuders (Stevanovic et al., 2015), and can account for light absorption enhancement due to "lensing effect", if absorption enhancement is considered independent of wavelength (Liu et al., 2015). The equations supporting this would be:

$$b^\lambda_{abs,IPN} = b^\lambda_{abs,BC} \cdot E^\lambda_{MAC_{BC}}(Lens) + b^\lambda_{abs,BrC}$$

Where, $E^\lambda_{MAC_{BC}}(Lens)$ is the coating related absorption enhancement, and $b^\lambda_{abs,BrC}$ (OA absorption) is zero at 1047 nm. The absorption coefficient measured at 1047 nm will account for the coating related absorption enhancement and this is what is extrapolated to the lower wavelength, giving us a representative value for BC absorption at those wavelengths.

**L142: Was a sonicator used? It would be surprising to find out that the samples were not sonicated during extraction.**

**The authors should report the extraction efficiency for the water solvent, as they can do this from the WSOC and OC measurements. What fraction of OA was extracted? It presumably must be small, or the correction factor for water versus methanol would not be all that large.**

No, a sonicator was not used during extraction due to concerns of mechanically dislodging BC from the filters (Phillips and Smith, 2017). We had initially thought of sending extracted and unextracted filters for EC/OC analysis to characterize extraction efficiency and determine mass absorption efficiencies but did not go through with it due to constraints with shipping filters. For justifying the validity of extractions without sonication, we would like to point the reviewer to work by Cheng et al. (2016), where they've performed methanol extractions without sonication and concluded that most organics were extracted. We allowed for longer dissolution times than Cheng et al. (2016), though they had larger solvent volumes for extraction.

We conducted the experiments in two sets, the first included water extraction along with TOC analysis of the extracts and the second set of experiments included extraction using all the plotted solvents. Data from the first set of experiments was used for the Mie calculations and as proof of concept that the extraction technique works and gives reasonable absorbance. The fraction of OA extracted by water had a broad range with values varying from 32-74% which are close to those observed by Chen and Bond (2010) for primary organic emissions.

**L178: "OC/TC ratios were assumed constant. . .". Was this assumption tested in any way?**

Yes, the assumption was tested by conducting experiments where two or more filters in a given burn were sent for EC/OC analysis. The EC/OC values were consistent and within error for all fuels except for Douglas fir. The instability for douglas fir emissions was noted early on as it was not possible to achieve a stable absorption coefficient signal for burns using this fuel. Consequently, douglas fir emissions were not extracted in any solvent, however data from these burns were used as points in the SSA v/s OC/TC plots. Table S2 now details the number and purpose of each filter collected in a given burn.

**L198: For the blacker samples (lower OC/TC), the BC will absolutely impact the retrieval of the real refractive index. How was this accounted for?**

The Mie calculations were performed for samples with OC/TC values of 1 to avoid assumptions for separating BC and OC size distributions from SMPS data. Hence, we did not have problems in retrieving real refractive indices. The retrieved real index for the sage burns was $1.61 \pm 0.12$ which is a reasonable estimate for OA emissions (Dinar et al., 2008; Sumlin et al., 2018).

**L212: The Cheng et al. reference is to a computational study. As much as those can shed insights, I suggest using an observational or lab study to make the case of the value for the SSA for BC. With the exception of some recent results from NIST (Radney et al., 2014, ES&T), I think that most experimental studies suggest lower values than stated here are possible.**

We have added a range of values for the expected SSA and made variations to the citations while also mentioning that SSA is highly sensitive to monomer size.

**L222: Presumably this power law was arbitrarily chosen? Were other forms explored? What is the predictive power of this, especially at high OC/TC? See the above comment about the data seeming to look more like a step function, than a power law.**

See response to the first comment where we explain the reason for using a power law over the step function. The predictive power of these parametrizations is not high, but better than equivalent step function counterparts. As these parametrizations are not predicting correction factors, we believe the low predictive powers for the parametrizations should not be a concern. The parametrizations in the updates manuscript are only provided to give some form of mathematical visualization to the data.

**L231: These are not "fluctuations." They are simply uncertainties. However, the error bars reported do not seem to reflect these uncertainties properly. There is no notable decrease in the size of the error bars below/above the thresholds identified. This begs the question, how were the uncertainties determined? The currently reported uncertainties in Fig. 2 are clearly underestimated, based on the potential for a 200% bias. Ultimately, the uncertainty is likely a direct function of the OC/TC, since the BC contribution will be larger when this ratio is smaller, and thus it will become increasingly difficult to separate OC from BC contributions. More than that, any uncertainty in the AAE will create a systematic, but OC/TC-dependent, bias in the key ratio determined here. I think that these issues need to be discussed in much greater detail.**

We have modified the paragraph to represent the importance of uncertainties due to AÅE. The error bars in Fig. 2 and Fig. 4 do not contain uncertainties associated with AÅE as is mentioned in the text. This was when we were presenting our results as correction factors instead of biases and had hence separated out "correction factors" with high errors from those with relatively low errors for potential use in different systems. We have revised this and added errors associated with AÅE uncertainties to our plots. The revised manuscript uses Monte Carlo simulations to estimate errors by assuming each value follows a normal probability distribution with certain standard deviations and have calculated errors based on that. We have also expanded our discussion on the role of OC/TC in these uncertainties.

**L242: This statement by the authors, that if they translate data from another study to the parameter space used here (OC/TC) they find different results, suggests that the premise of this study might be flawed. This suggests that the results here might not translate well to other settings, and thus the fit function determined in Fig. 2 is not robust beyond the current study. The authors need to address the issue of how robust they expect their parameterization to be, and how extensible to other systems. Also, the fact that the current study and a previous study disagree so much seems to limit the statement on L253 that the SSA can be predicted from the OC/TC. The prediction from the fits in this study may simply not be robust.**

We have added to the section by using our fit to predict values from another study and have suggested reasons for possible mismatch between the fits obtained from our study and those by Pokhrel et al., (2016). We hypothesize that the main reason for the difference in fit parameters

could be the result of potential difference in EC/OC ratios obtained by different instruments even though the same thermal protocol was used in both studies (Panteliadis et al., 2015). The reason for comparison with Pokhrel et al., was to show that even though the parameters for the fits were different, both studies obtained linear correlations between SSA and OC/TC (or EC/TC) indicating that some form of systematic linear trend between the two parameters exists.

**L279: The authors need to clarify how specifically an OC/TC dependence of the OA absorptivity explains the apparent difference in methanol versus water.**

We hypothesize that the temperature for burns which lead to high EC fractions in the aerosol also lead to the release of larger amounts of ELVOCs (Saleh et al., 2014) or polycyclic aromatic hydrocarbons (PAHs) which might not be completely extractable by water but are more readily extracted by methanol. These compounds have been shown to have high absorption efficiencies which could lead to the observed difference in absorption even though extraction efficiencies might not be drastically different. Thus, absorption coefficients for WSOC are lower than MSOC and corresponding AÅE values are higher as these compounds are expected to absorb light at longer wavelengths as well . The text has been modified to represent this.

**L284: Are the authors deriving their conclusion that the Zhang et al. (2013) results support the findings here from the following sentence in Zhang: "The water-insoluble BrC, calculated as the difference between methanol- and water-extracted BrC, exhibited a tighter correlation with ambient EC concentrations (r2 = 0.81, Figure 5b) than water-soluble BrC (r2 = 0.40), suggesting that the water-insoluble BrC components and EC have similar sources (e.g., incomplete combustion from vehicle emissions and wood burning)." As best I can tell, this is the only sentence that might connect. But I am skeptical of the relevance, since in the Zhang case the distinction is largely between primary and secondary OA, not different types of primary OA. I suggest that the authors' argument needs to be strengthened if it is to be kept.**

We have removed the Zhang et al. (2013) citation and modified our argument for the increase in $b_{abs,OA}/b_{abs,bulk}$ with increasing EC fractions of our aerosol. The argument now reads:

"The differences in the magnitudes of the correction factors between acetone/methanol extracts and water extracts increase as the EC composition of the aerosols increases. An increase in extremely low volatility organic compounds (ELVOCs) with increasing EC/OC ratios was observed by Saleh et al. (2014) and we hypothesize that these ELVOCs which have high mass absorption efficiencies (Saleh et al., 2014; Di Lorenzo and Young 2015) could have a lower solubility in water than methanol or acetone which would explain the increasing difference in $b_{abs,OA}/b_{abs,sol}$ values between water and methanol/acetone extracts."

**L294: It is not clear how the authors come to the conclusion that "higher molecular weight" compounds are responsible here. This seems like speculation and should be posed as such.**

We have changed our hypothesis for lower AÅE of methanol to as shown below:

"Experiments by Zhang et al., (2013) observed absorption by polycyclic aromatic hydrocarbons (PAHs) at longer wavelengths close to the visible region. Organic compounds such as methanol

have a higher extraction efficiency for these compounds than water leading to higher absorption by methanol extracts at longer wavelengths which results in lower AÅE (Zhang et al., 2013)."

**L288 and AAE discussion: The authors do not present an error analysis here, and the uncertainties on the lower OC/TC samples will be very large if propagated appropriately. This is especially important for any of the conclusions reached regarding comparison between particle and solution-phase differences.**

We have propagated the uncertainties and included corresponding errors to our results. The conclusions have been modified to reflect this. We used a Monte Carlo simulation to estimate the mean and errors for the particle phase.

**L300: The authors state that the particle phase AAE are "close to" those in the solution phase at high OC/TC. But then in the next sentence they state that the particle phase and solution-phase "deviate significantly." These seem contradictory. I actually do like this general aspect of the analysis (especially at higher OC/TC, where uncertainties from extrapolation are smaller). But, without a robust error analysis and a more quantitative discussion of the comparison I don't think the authors can arrive at their conclusions. At minimum, there should be something like a t-test to check for statistical differences between the particle- and solution-phase AAE values. I suspect that if the authors include all data (not just dung) they will find that the methods give statistically indistinguishable AAE values.**

We carried out a t-test for differences in water and particle phase AÅE for 12 samples with OC/TC ratios $\geq 0.90$ and differences are significant with p values $< 0.05$. The difference is however not statistically significant for water when all samples are compared (p $\approx$ 0.15, 18 data points) excluding those with OC/TC ratios $< 0.7$ as these data points have high errors in AÅE. The differences are extremely statistically  significant when comparing all samples with OC/TC ratios $\geq 0.95$ (11 data points) and $> 0.7$ (18 data points) with p values $< 0.0005$ when comparing particle phase AÅE with acetone and methanol.

**Reviewer 2:**

**Lines 114/116: The ratio of OC/TC were reported of ranges 0.55-1 but figure 3 (line 489) shows that ratios observed are only in the specific data ranges such as 0.55, 0.6, and 0.7. 0.8, and 1. It is not clear why the OC/TC data of ratios in between those ranges, such as of ratios 0.65, 0.75, and 0.85, were not observed and not reported. Are these ratios were rounded? And also, there is no excess data of ratios about 0.55 and 0.6 which contribute for analysis. I have the impression that the significant correlation, ~0.95 of SSA vs OC/TC is mainly driven by some outliers. I strongly recommend presenting error analysis on these data sets**

The different OC/TC mass ratios were obtained by isolating flaming/smoldering combustion phases or by allowing emissions from the two phases to mix together. This was done by prematurely extinguishing the flame and sampling single or mixed phase emissions. This did not allow us to exactly control the values of OC/TC ratios that were generated. The ratios were not rounded and the fact that we could not observe values such as 0.65, 0.75 and 0.85 was just a matter of probability. We have made new plots for Figure 3 which represent the errors in the OC/TC

ratios and SSA along with 95% confidence intervals for our fits. We are now reporting the RMSE for the fit instead of the r values thereby removing any effects of outlier points. We have also modified the regression from a least square method to an error-in-variables model (orthogonal distance regression) to account for uncertainties in the OC/TC ratio.

**Lines 124/126: Why was SMPS not used for all of the experiment? Please clarify this. Also, give a reference or a brief explanation of how the geometric mean size distribution was determined?**

The experiments were conducted in two batches, once by only performing water extractions and the other where organics were extracted in water, methanol and acetone. The SMPS measurements along with the TOC analysis were performed on samples collected during the first batch of experiments only and data from these were used in the Mie calculations. We have added text in the manuscript to indicate this difference. The geometric mean was evaluated by the TSI software using the general formula for geometric mean size. We did not find it necessary to include a description for calculating the geometric mean as it is not relevant to the following discussion.

**Line 151: There are no references or derivation of mathematical equations used in the manuscript, for example, Eqn.1, babs, bulk (λ). Please provide the references or derivations to support the validity of the mathematical equations used in the manuscript.**

Citations have been added to corresponding references for the different equations used throughout the manuscript.

**Line 154: Explain why absorbance at a given wavelength is normalized to the absorbance at 700nm.**

We have provided an explanation which states that the absorbance is normalized to values at 700 nm to account for any signal drift within the UV-Vis spectrophotometer signal.

**Lines 183/185: What is the range of assumptions made along with Mie theory, as stated in the text? Reference is recommended to include for determining the imaginary complex refractive index**

Mie Theory assumes that the particles are uniform and isotropic with a spherical shape, and that these particles are fully illuminated in an infinite dielectric medium. In addition to this, the real part of the complex refractive index needs to be assumed as well. We have modified the sentence to read:

"…absorption, using Mie Theory along with assumptions regarding the shape of the particles and the real part of its complex refractive index."

A reference to the equation has been added.

**Line 223: How were the RMSE values calculated? Please include the reference/formula or name of software which was used to get RMSE values in Table 1, such as excel MATLAB, or Igor Pro.**

The RMSE values were calculated using Excel. We have added a sentence in the manuscript to reflect this.

**Lines 223/224: Please add a line to justify the impact of BC AAE on conversion factor for particles with SSA smaller than 0.7 at 375nm and smaller than 0.825 at 405 nm.**

The next two sentences in the manuscript justify the impact of BC AÅE on $b_{abs,OA}/b_{abs,bulk}$. The uncertainty in BC AÅE could lead to uncertainties close to 200% at smaller SSA values. We have added a sentence explaining that the increase in uncertainty would be due to an increase in BC mass concentrations with decreasing SSA.

**Line 225: What is sensitivity analysis as stated in the text? Please explain briefly**

The sensitivity analysis is performed to observe how sensitive the measurements are to variations in certain assumed variables ($AÅE_{BC}$ here). To perform the analysis, the value of $AÅE_{BC}$ was varied from 0.85 to 1.1 and we noted the corresponding change in $b_{abs,OA}/b_{abs,bulk}$. The differences were compared to values where $AÅE_{BC}$ is 1 and the differences are indicative of how sensitive the result is to given variable.

**Line241/242: Briefly describe why the two different studies, Pokhrel et al. (2016) and the current studies give different slopes and intercepts of the resulting fits?**

We hypothesize that the main reason for the difference in fit parameters could be the result of potential difference in EC/OC ratios obtained by different instruments even though the same thermal protocol was used in both studies (Panteliadis et al., 2015). A section where we compare our fit to another study has also been added to this section.

**Line 276: What does SI represent for?**

SI stands for Supplementary Information and has been expanded as so in the manuscript.

**Lines 292/295: Briefly explain why the higher molecular weight compounds absorb more light?**

We have changed our hypothesis for differences in AÅE between water and methanol extracts and have added text explaining that this may be due to increased extraction of polycyclic aromatic hydrocarbons (PAHs) by methanol. These PAHs absorb more light at longer wavelengths which would result in lower AÅE values for methanol when compared to water.

**Lines 296: AAE values for OA are significantly high with wide ranges of 4.4 -14.61. How are these values related to wavelengths? Please provide some references, if there are any, to support these values**

Few studies look at AÅE of OA in the UV range making it difficult to find relevant citations. However, our AÅE values are similar to those observed by Chen and Bond 2010 for OC extracted in methanol and water. Pokhrel at al. 2016 observed AÅE ranging from 3.7 to 10.4 for wavelengths 405/532/660 which are close to the values observed in this study.

**Lines 296/297: It is reported that overall AAE for OA decreases with increased EC. Please add a graph/or a brief note to show AAE for OA measurements with the concentration of EC.**

We have removed this argument from the manuscript.

**Line 342: Authors' names are not clearly reported: RKC, SB, WMH, NS, AP, are not previously reported with these names in the authors' list. I think it is not relevant to include author contributions in the manuscript once a list of authors is reported.**

The author contribution list was added as it is a requirement for publication in ACP.

**Reviewer 3:**

**(1) The authors effectively define the in-situ shortwave absorption coefficient for organic aerosol, babs,OA, as the excess of the PAS-measured total over the BC contribution. They extrapolate the necessary shortwave BC value from a longwave PAS measurement, via an assumed unit AAE. This BC contribution is the proverbial 'elephant in the room', appearing nowhere in the results but providing essential context for their interpretation. What are the relative contributions of OA to total absorption at short wavelengths, and how do they vary with fuel type and burn conditions? Extended AAEs (AAE405-1047 or AAE375-1047), from which a curious reader could derive an answer, are nowhere indicated. A related question is how longwave absorption and TOR EC relate to each other, since they are independent proxies for the same BC. Figure 3 plots only ratios against ratios, shortwave bscat/(bscat+babs) from the in-situ IPN measurement against OC/TC from the filter TOR analysis. It would seem at least equally instructive to compare the concentration values directly, longwave babs against TOR EC**

The relative contribution of OA absorption ranged from 23 to 97% at 375 nm and from 7 to 96% at 405 nm. The fraction of OA absorption decreases with increasing EC fraction of the aerosol. In Figure 3 we plotted SSA with OC/TC to compare the results with similar trends observed by Pokhrel et al. (2016). TOR EC was converted to EC mass concentrations based on the sampled volume and $b_{abs}$ at 1047 nm was plotted against the EC mass concentration. The plot showed an increase in the $b_{abs}$ with increasing EC mass fraction as expected, however, the scatter was high with a few outlier points at high EC mass concentrations. We have added a Table with extended AÅEs, both AÅE$_{375-1047}$ and AÅE$_{405-1047}$ in the Supplementary Information, but believe that adding the plot for $b_{abs}(1047)$ against TOR EC might not be as helpful due to the lower correlation as compared to the trend between SSA and OC/TC.

**(2) Two SVOC denuders sit between the burn chamber and the holding tank (Figure 1), but we can expect some phase re-equilibration to occur within the holding tank before samples are drawn. Adsorption of re-volatilized organic species by the quartz sampling filters will then generate artifacts in the TOR and extraction measurements that are not present in the IPN optics measurements. Did the authors collect and analyze quartz blanks to quantify**

**these artifacts, using a (non-adsorbing) PTFE filter between the smoke-filled holding tank and sampling port to exclude the particle phase?**

We did not account for phase partitioning of the of the SVOCs within the holding tank. We have modified the text to indicate that this might lead to positive artifacts in the OC measurements and could lead to an increase in solution phase absorption coefficients. However, the contribution to absorption by SVOCs is small compared to the non-volatile organics (Chen and Bond 2010) and should not affect our measurements by a lot with adsorption artifacts contributing an OC error of 1-3% (Pokhrel et al., 2016).

**(3) It is hard to relate and reconcile the experimental data shown or listed in the different figures and tables. Not all IPN- and TOR-characterized burn samples were filter-extracted for OA, and not all water extracts were analyzed for TOC. It would be helpful to make these experimental layers clearer to the reader, along with some indication of criteria for inclusion/exclusion. For example, Table S1 lists 53 filters collected from 28 burns. I infer that this (53) counts just the filters consumed for TOR OC/TC analysis, each paired with another filter collected for extraction (lines 133-134). (If the 7 filters from dung burns instead represented all TOR and extraction filters together, then we would have at most 3 pairs yielding complete records, contra the 4 observations listed in Table 4.) It appears from Table 4 that only 21/53 ~ 40% of the other filters were selected for quartering and extraction. Does Figure 3 show all 53 observations from Table S1? Do Figures 2 and 4 show those 21 observations from Table 4? It would be helpful to be told the total number of observations appearing in each figure and table.**

We conducted the experiments in two sets, the first included water extraction along with TOC analysis of the extracts and the second set of experiments included extraction using all the plotted solvents. Data from the first set of experiments was used for Mie calculations and as proof of concept that the extraction technique works and gives reasonable absorbance. Data plotted in all graphs and the AÅE table were analyzed during the second set of experiments. We thank the reviewer for pointing out the confusion regarding this and have added Table S2 which details how each filter was used and corresponding OC/TC ratios. We have also added the number of data points used in each plot. As for dung, all dung burns had no visible flaming combustion phase and smoldered throughout the combustion process and the one odd sample without a corresponding OC/TC filter was just assumed to have an OC/TC value of 1. Removing this data point does not affect the fit coefficients and conclusions by a lot and we believe that assuming the dung emissions to be purely organic is justified.

**(4) Is it true that the TOR analyses were performed AT Sunset Laboratories (line 238), and not locally with an instrument manufactured BY Sunset Laboratories?**

Yes, the TOR analysis was performed at Sunset Laboratories.

**(5) In addition to flowrate and sampling time (line 133), filter area is a relevant experimental factor and should be specified.**

We agree with the reviewer, a 47 mm  diameter quartz fiber filter was used for sampling and then quartered and used for analysis. The same has been added to the text as well in Section 2.2.1 "Quartz filters (Pallflex Tissuquartz, 47 mm diameter) collected during sampling were split…"

**Reviewer 4:**

**Given the comments already made by the other three anonymous reviewers, I will refrain from repeating what they have stated. I agree with Reviewer #1's assessment that the conclusions from this work are not sufficiently general to be of use for correcting bulk, solvent-based absorption measurements. As the other reviewers have pointed out, the measured correction factors incorporate not just geometric differences in bulk and particle absorption but also solvent- and constituent-specific factors, including solubility. And, there are correction factors measured at nearly identical SSA or OC/TC values that differ by factors of 2-3 (Figures 2 and 4) – such scatter is too great to draw a meaningful conclusion about the dependence of the correction factors on SSA or OC/TC ratio. It appears as if no dependence, i.e. a horizontal line, would describe the trends about as well as the arbitrarily-chosen power law function.**

Based on comments by the other reviewers, we have made substantial changes to the manuscript and modified the conclusion as well and are not purporting our findings as correction factors any more. We had considered the use of a step function over a power law, the reason for selecting the power law fit was to keep the equation consistent through all regressions. It would appear that a step function is better than the power law for the water extracts, but the gradual slope with OC/TC ratios and SSA is more prominent in the methanol and acetone extracts. In addition to this, the root mean square error (RMSE) values for the power law fit were consistently lower than their step function counterparts. The power law fit can also mimic a step function with a steeper slope and can also have a curve with a gradual slope depending on the value of the power law exponent.

**In short, the main conclusion from this study is that there are different correction factors for water and methanol/acetone with water extracting less absorption than the other solvents. This conclusion is not new and may not be general to other types of absorbing organic aerosols or even other types of biomass burning aerosols. What is more, the extent of scatter makes potential use of these factors problematic. Hence, the factors measured here are not broadly applicable. Furthermore, the purported dependence of these factors on SSA or OC/TC is overstated making that conclusion suspect as well.**

The conclusions have been changed.

**References:**

Chen, Y., & Bond, T. (2010). Light absorption by organic carbon from wood combustion. Atmospheric Chemistry and Physics, 10(4), 1773-1787.

Cheng, Y., He, K.-b., Du, Z.-y., Engling, G., Liu, J.-m., Ma, Y.-l., . . . Weber, R. J. (2016). The characteristics of brown carbon aerosol during winter in Beijing. Atmospheric environment, 127, 355-364.

Dinar, E., Riziq, A. A., Spindler, C., Erlick, C., Kiss, G., & Rudich, Y. (2008). The complex refractive index of atmospheric and model humic-like substances (HULIS) retrieved by a cavity ring down aerosol spectrometer (CRD-AS). Faraday discussions, 137, 279-295.

Liu, S., Aiken, A. C., Gorkowski, K., Dubey, M. K., Cappa, C. D., Williams, L. R., . . . Chhabra, P. S. (2015). Enhanced light absorption by mixed source black and brown carbon particles in UK winter. Nature communications, 6, 8435.

Panteliadis, P., Hafkenscheid, T., Cary, B., Diapouli, E., Fischer, A., Favez, O., . . . Vecchi, R. (2015). ECOC comparison exercise with identical thermal protocols after temperature offset correction: instrument diagnostics by in-depth evaluation of operational parameters.

Phillips, S. M., & Smith, G. D. (2017). Spectroscopic comparison of water-and methanol-soluble brown carbon particulate matter. Aerosol Science and Technology, 51(9), 1113-1121.

Pokhrel, R. P., Wagner, N. L., Langridge, J. M., Lack, D. A., Jayarathne, T., Stone, E. A., . . . Murphy, S. M. (2016). Parameterization of single-scattering albedo (SSA) and absorption Ångström exponent (AAE) with EC/OC for aerosol emissions from biomass burning. Atmospheric Chemistry and Physics, 16(15), 9549-9561.

Saleh, R., Robinson, E. S., Tkacik, D. S., Ahern, A. T., Liu, S., Aiken, A. C., . . . Yokelson, R. J. (2014). Brownness of organics in aerosols from biomass burning linked to their black carbon content. Nature Geoscience, 7(9), 647.

Stevanovic, S., Miljevic, B., Madl, P., Clifford, S., & Ristovski, Z. (2015). Characterisation of a commercially available thermodenuder and diffusion drier for ultrafine particles losses. Aerosol and Air Quality Research, 15(1), 357-363 (i.

Sumlin, B. J., Heinson, W. R., & Chakrabarty, R. K. (2018). Retrieving the aerosol complex refractive index using PyMieScatt: A Mie computational package with visualization capabilities. Journal of Quantitative Spectroscopy and Radiative Transfer, 205, 127-134.

---

## Referee Report (RR1)

Measuring Light Absorption by Primary Organic Aerosols: Optical Artifacts in Traditional Solvent Extraction-Based Methods

Shetty et al's new version of the manuscript is much clearer and addressed the major concerns highlighted in the reviewer comments. However, I still see some minor errors and contents that need to be addressed in the manuscript. In the manuscript, the symbols in each equation and in the text should be written in the same format such as in the Microsoft Equation format. It seems that, in the text, symbols are written in Microsoft font and not in the Microsoft equation format. I recommend using a comma between two lambdas (in line 196: $\lambda_1$, $\lambda_2$). In page 319, add a line by giving a reference that can be referred to obtain 0.39 for BC using parametrization at OC/TC ratio of 0 (Line 319) or add a line to support it. Also, explain is it possible to observe OC/TC as zero (line 319)? In lines 353-354, Fig. 5 does not tell anything about EC fractions>0.25 and typical BB as stated in the text. Add a line showing how it is associated with Fig. 5. Section 3.4, the title is recommended to rewrite such as Scaling Factors Based on Mie calculations. In line 408-409, it seems ambiguity of suggesting researchers avoid such scaling factors for determining OA absorption without exact knowledge of OC extraction efficiencies and particle size distributions. I suggest to authors to present data or references to support how knowledge of OC extraction efficiencies and particle size distributions will convince of using scaling factors for determining OA absorption.

---

## Author Response (AR2)

We thank all the reviewers (including the editor) for their useful suggestions and insight. We have made appropriate changes in our manuscript to incorporate the reviewer comments. Based on verbal conversations with experts in the field, we have also made a small change to the title replacing "primary organic aerosol" with "freshly emitted organic aerosol" as we are not certain that no secondary processes occur within the volume of our chamber leading to sampling of some secondary aerosols as well. Below are our responses to concerns brought up by the reviewers with reviewer comments in **boldface** followed by our reply.

**Review:**

**Dr. Sergey Nizkorodov**

**Dear authors. As reviewers pointed out, the revised version has improved considerably. Please address the remaining points raised by reviewers #1 and #2 of the revised manuscript. In addition, I have a few editorial review notes below, which may also need to be addressed before the final acceptance.**

**Line 157: a space is needed between l and min**

The space has been added

**Line 170: there is another important artifact of changing the chemical composition by sonochemistry, as described in [Mutzel, A.; Rodigast, M.; Iinuma, Y.; Böge, O.; Herrmann, H., An improved method for the quantification of SOA bound peroxides. Atmos. Environ. 2013, 67, (0), 365-369.] You might want to cite it.**

The citation has been added along with the line: "…and to avoid changes in chemical composition caused by acoustic cavitation (Mutzel et al., 2012)". Thank you for providing this citation.

**Line 173: impurities -> suspended particles (because filtration does not remove soluble impurities)**

Impurities has been changed to suspended particles

**Line 182: this effectively sets the absorption coefficient at 700 nm to zero. Please discuss in the paper whether it was actually the case for your samples (in many cases absorbance at 700 nm is indeed negligible). It might be more precise to subtract an average absorbance over a wavelength interval where you expect no absorption, such as 700-750 nm as opposed to subtracting a single value, which is prone to more uncertainty. I do not suggest you should redo all of your calculations, just a suggestion for the future.**

We now mention that the absorbance at 700 nm was negligible and close to zero for our samples. The value of 700 nm was selected based on the convention set by previous studies. However, we agree that an average value over the long wavelength range would be more appropriate for future calculations. The authors thank you for the suggestion.

**Line 189: please fix grammar in this sentence**

The sentence now reads: "It was assumed that all the absorption at 1047 nm could be attributed to BC aerosol (Bahadur et al., 2012)"

**Line 235: I think conversion of WSOC into mass concentration of dissolved organics requires an OM/OC ratio. What was the assumed value?**

The mass concentrations mentioned here are OC mass concentrations. The sentence now reads: "In the given study, the OC mass concentration was measured…" to get rid of any confusion. We have also added a line stating that the obtained refractive index is representative of OC and not OM: "It is important to note that $k$ values obtained using this method will represent optical characteristics of OC mass and not total organic mass."

**Table 4: there are too many significant digits reported in AAE values. There is no reason to report uncertainties with more than 1-2 significant digits. The entry 13.74+/-2.27 would be more faithfully represented as 14+/-2 or 2 13.7+/-2.3. Similar considerations apply to numbers reported in a few other tables as well as text.**

The number of significant figures has been reduced for AAE and fit coefficients. These have been modified throughout the text as well.

**Several references have an incomplete list of authors, for example, Akagi et al. (2011), Arnott et al. (2003). I think your reference manager software is cutting the list. ACP lists all authors to the best of my knowledge, so it needs to be fixed.**

**Also, fix page numbers in references Arnott et al. (2003), Bahadur et al. (2012), Kirchstetter et al. (2004), Saleh et al. (2015), Sun et la. (2007), Zhang et al. (2008). I am guessing your reference library is missing these page numbers and formatting the references incorrectly. It is a common problem form AGU journals.**

**Panteliadis et l. (2015) reference is missing the journal**

All the references have been fixed according to the ACP standard, thank you for pointing these out.

**Anonymous Referee #2:**

Shetty et al's new version of the manuscript is much clearer and addressed the major concerns highlighted in the reviewer comments. However, I still see some minor errors and contents that need to be addressed in the manuscript.

**In the manuscript, the symbols in each equation and in the text should be written in the same format such as in the Microsoft Equation format. It seems that, in the text, symbols are written in Microsoft font and not in the Microsoft equation format.**

The symbols have been modified to the Microsoft equation format in both the text as well as in equations

**I recommend using a comma between two lambdas (in line 196: $\lambda 1$, $\lambda 2$).**

The "and" was replaced with a comma.

**In page 319, add a line by giving a reference that can be referred to obtain 0.39 for BC using parametrization at OC/TC ratio of 0 (Line 319) or add a line to support it. Also, explain is it possible to observe OC/TC as zero (line 319)?**

We have added references stating that observations from most studies give lower SSA, but those by Radney et al. (2014) are closer to values obtained from our parametrizations. The line reads: "Most observations for soot SSA are lower than those predicted by our 405 nm parametrizations (Bond et al., 2013, Schnaiter at al., 2003) with our projections being closer to SSA observed by Radney et al. (2014)."

We have also changed the sentence from OC/TC of 0 to: "…SSA value of 0.39 for pure EC obtained using our parametrization." bypassing the need for explaining observations having OC/TC ratios of zero.

**In lines 353-354, Fig. 5 does not tell anything about EC fractions>0.25 and typical BB as stated in the text. Add a line showing how it is associated with Fig. 5.**

EC fractions > 0.25 are representative of values with OC/TC ratios < 0.75 as depicted in Fig. 5. We have changed "EC fractions > 0.25" to "EC/TC ratios > 0.25". While Fig. 5 does not say anything about typical BB, the previous line references typical OC/TC values observed for lab- and field-based BB experiments justifying the need to exclude these points from our analysis.

**Section 3.4, the title is recommended to rewrite such as Scaling Factors Based on Mie calculations**.

The title of Section 3.4 was changed.

**In line 408-409, it seems ambiguity of suggesting researchers avoid such scaling factors for determining OA absorption without exact knowledge of OC extraction efficiencies and particle size distributions. I suggest to authors to present data or references to support how knowledge of OC extraction efficiencies and particle size distributions will convince of using scaling factors for determining OA absorption.**

We thank the reviewer for pointing out the apparent ambiguity in the sentence. We have changed the sentence to "We recommend future studies to use caution and judgement when using *a priori* scaling factors for determining OA absorption using solvent extraction techniques."

**Anonymous Referee #5:**

**1) The trends in Figure 2 and Figure 5 seem to be unrealistic, or at least inconsistent with the argument (Line 358-365) that extraction efficiency increases with increasing OC/TC (and thus, SSA). Specifically, in panel (b), how do the authors explain the sharp drop in babs,OA/babs,sol to values close to zero at high OC/TC and SSA? One would expect babs,OA/babs,sol to approach a constant value (due only to particle size effects) with increasing extraction efficiency, so one should expect an asymptotic behavior of babs,OA/babs,sol at large OC/TC and SSA values, opposite to what's observed. It seems to me that the observed trends could be partially an artifact of the way babs,OA is calculated as the difference between the measured babs,tot and a calculated babs,BC (section 2.2.2). In any case, a discussion of the meaning of the trends observed at large SSA values should be added to section 3.1.**

The argument for decreasing extraction efficiencies in water was used to explain the increasing difference in absorption bias between water and methanol extracts. In panel (b) for methanol and acetone extracts, rather than being close to zero, these values were close to 0.6 for some dung samples which are similar to theoretical predictions by Sun et al. (2007) for particles much smaller than the wavelength of light; but we were not confident of justifying these observations based on this argument as our size distributions were comparable to the light wavelength of 405 nm. We have added a line to acknowledge these observations in Section 3.1.

Based on our observations, we believe that the extraction efficiency would not decrease indefinitely but tend towards a constant value above a given EC/OC fraction depending on the type of organics released, mimicking an exponential function comparable to observed trends in biomass burning OA wavelength dependence with EC/OC ratios (Saleh et al., 2014) and OA refractive indices (Saleh et al., 2018). This would lead to the bias approaching a constant value (due only to particle size effects) with decreasing OC/TC ratios and in turn the aerosol SSA. The text in the revised manuscript has been modified to better reflect this fact. The authors thank the reviewer for pointing the ambiguity in our explanation.

Based on sound absorption theories, we do not believe that our observations are an artifact of the calculations. For example, the widely accepted band-gap model (Moosmüller et al., 2011) has been used to support a BC AÅE of 1; this value has been used by the community to extrapolate spectral absorption coefficients from longer to shorter wavelengths across the near-UV-Vis-NIR solar spectrum. The ideal nature of the band gap model could have deviations, leading to observed BC AÅE ranging from 0.85 to 1.1 as found in other studies (Lack et al., 2008; Bergstrom et al., 2007; Lan et al., 2013), and these extremes have been taken into consideration in our study for sensitivity calculations.

**2) Along the same lines, the data, especially the part of the space that has the sharp change (i.e. SSA > 0.9), should be further explored. For example, the data in Figure 2 could be plotted with the points color-coded by fuel type and with different scales on the y-axis to better zoom in the changes at SSA > 0.9 to see if the variability that is not explained by SSA**

**(i.e. where the changes are very steep) could be explained by fuel type. The same applies to OC/TC.**

The authors could not find any variability that could be clearly explained by the fuel type, and no interesting or noticeable findings were observed with exploring the data near the sharp change. Color coding the points based on fuel type was a helpful suggestion, however doing so made the figures busy and hard to follow. We have incorporated the reviewer's suggestion by modifying the figures with different markers for each fuel and color coded based on the solvent used for extraction. As the effects of fuel type were not apparent, these have not been discussed in the manuscript.

**3) Line 275 – 281: the choice of SSA = 0.7 and 0.825 as a cutoff between low uncertainty and high uncertainty ranges seems arbitrary. What constitutes high uncertainty? Also, in Figure 2, similar size error bars (especially in panel b) seems to exist on both sides of the cutoff lines.**

While the error bars were calculated using the Monte Carlo simulations, the cutoff for high and low uncertainty was based on sensitivity to range of BC AÅE observed by past researchers. For samples below the cutoff, a change in BC AÅE from 0.8 to 1.1 led to drastic changes in the $b_{abs,OA}/b_{abs,sol}$ values with uncertainties greater than 30%, and in one case exceeding 200%. The high uncertainties were a direct result of the high absorption coefficient at 1047 nm for the given samples. We have modified the text to indicate what constitutes high uncertainty in our analysis. As the error bars were calculated with the Monte-Carlo simulations, the errors seem to be similar on both sides of the cutoff line, but the points above the cutoff have slightly smaller sensitivity to BC AÅE than the points below it.

**4) The purpose and value of the Mie calculations in section 3.4 are not clear. It reads: previous studies have used scaling factors of 2. This study found a scaling factor of 2. In conclusion, a scaling factor of 2 should not be used because of reasons stated in the previous sections. Then why do the Mie calculations to begin with? The cautionary statement about the non-universality of the scaling factor of 2 should be inserted in section 3.1.**

**The presentation of the results could be reformulated in a more informative way. For the 3 experiments reported in section 3.2, the authors can compare babs,OA, babs,sol, and babs,Mie and use this comparison to determine the fractional contribution of particle size and extraction efficiency to the discrepancy.**

The Mie calculations were added to check the reproducibility of the conventionally used scaling factor. We found that the factor was reproducible for the samples that we tested; however, these values had errors while predicting particle phase absorption coefficients and pointing this out through our calculations was the purpose of this section. We have modified the text in this section to explain this better.

The extraction efficiency for the tested samples were close, with $61 \pm 2\%$ extraction of organics for the three samples. As the actual differences in the scaling factors is not significant, and the Mie Calculations were not too sensitive to the particle size distributions we did not find it helpful to compare $b_{abs,OA}$, $b_{abs,sol}$ and $b_{abs,Mie}$ separately. We have however modified the text to better explain the reason for this section

**5) Table 4: It would be more informative to present the data in Table 4 as scatter plot of AAE_OA vs AAE_sol, color coded with fuel type and with a 1:1 line.**

We tried depicting the data using a scatter plot along with error bars, but the final plots looked very busy and did not provide better insights than those obtained using the Table along with a t-test analysis. For this reason, we preferred to represent the data using the table rather than a scatter plot. We however thank the reviewer for their suggestion and have added a similar plot in the supplementary section representing particle phase AAE and AAE data from each solvent extract, having different markers based on fuel type.

**6) Table 5: with geometric mean diameter of 397 nm, the size distribution is expected to extend beyond the SMPS measurement window. It is not clear how this issue was addressed.**

The issue was resolved by extending the SMPS data using the equation for a lognormal function. A line has been added to the methods section 2.3 clarifying the same: "If size distributions extended over the SMPS measurement range, the data were extrapolated using a lognormal equation."

[revised manuscript text omitted]